# CABINET: Content Relevance based Noise Reduction for Table Question Answering

[*][+]**Sohan Patnaik**[1,2], [*][+]**Heril Changwal**[3], [*]**Milan Aggarwal**[1], **Sumit Bhatia**[1],
**Yaman Kumar Singla**[1], **& Balaji Krishnamurthy**[1]
[1]MDSR Lab, Adobe; [2]IIT Kharagpur; [3]IIT Roorkee
{sohanpatnaik106, changwalheril}@gmail.com
{milaggar, sumbhati, ykumar, kbalaji}@adobe.com

## ABSTRACT

Table understanding capability of Large Language Models (LLMs) has been extensively studied through the task of question-answering (QA) over tables. Typically, only a small part of the whole table is relevant to derive the answer for a given question. The irrelevant parts act as noise and are *distracting information*, resulting in sub-optimal performance due to the vulnerability of LLMs to noise. To mitigate this, we propose **CABINET** (**C**ontent Relev**A**nce-**B**ased No**I**se Reductio**N** for Tabl**E** Ques**T**ion-Answering) – a framework to enable LLMs to focus on relevant tabular data by suppressing extraneous information. CABINET comprises an Unsupervised Relevance Scorer (URS), trained differentially with the QA LLM, that weighs the table content based on its relevance to the input question before feeding it to the question-answering LLM (QA LLM). To further aid the relevance scorer, CABINET employs a weakly supervised module that generates a parsing statement describing the criteria of rows and columns relevant to the question and highlights the content of corresponding table cells. CABINET significantly outperforms various tabular LLM baselines, as well as GPT3-based in-context learning methods, is more robust to noise, maintains outperformance on tables of varying sizes, and establishes new SoTA performance on WikiTQ, FeTaQA, and WikiSQL datasets. We release our code and datasets here.

## 1 INTRODUCTION

Understanding tabular data through ML models has been extensively explored through various tasks such as question-answering (QA) (Chen et al., 2022; Cheng et al., 2022; Nan et al., 2022), fact-verification (Chen et al., 2020b; Wang et al., 2021; Aly et al., 2021), table-to-description generation (Chen et al., 2020a; Parikh et al., 2020; Chen et al., 2020c; Suadaa et al., 2021; Nan et al., 2021) and table grounded dialogue (Nakamura et al., 2022). Table QA has been studied with a specific focus as it allows to conveniently query the table in natural language to extract desired information. Large Language Models (LLMs), which have shown remarkable generalization on various Natural Language Processing (NLP) tasks, have also been used to reason over tables achieving impressive performance (Yu et al., 2021; Neeraja et al., 2021; Gu et al., 2022; Chen, 2023).

Tables contain information organized in rows and columns, and typical transformer-based LLMs such as BERT (Devlin et al., 2019), T5 (Raffel et al., 2020), and GPT (Brown et al., 2020) trained over unstructured natural language text using standard language modeling objectives do not account for the table structure and underlying compositionality of data (Yu et al., 2021). Many works on table understanding therefore, adapt LLMs for tables through joint learning over tabular and text content (Yin et al., 2020), pre-training on table semantic parsing (Liu et al., 2022; Jiang et al., 2022) and synthesizing template-based questions to improve reasoning skills over tables (Gu et al., 2022). Typically, only a small number of cells contain the information required to derive the answer for a question. The irrelevant tabular data acts as *distracting information* or noise, resulting in sub-optimal performance since LLMs are susceptible to noise in the input (Kumar et al., 2023; Chen et al., 2023a). Performance degradation is further amplified in large tables due to presence of even more data as illustrated in Figure 4 in Section 4.3.

---

[*]equal contribution; [+]work done during internship at Adobe Media and Data Science Research Lab

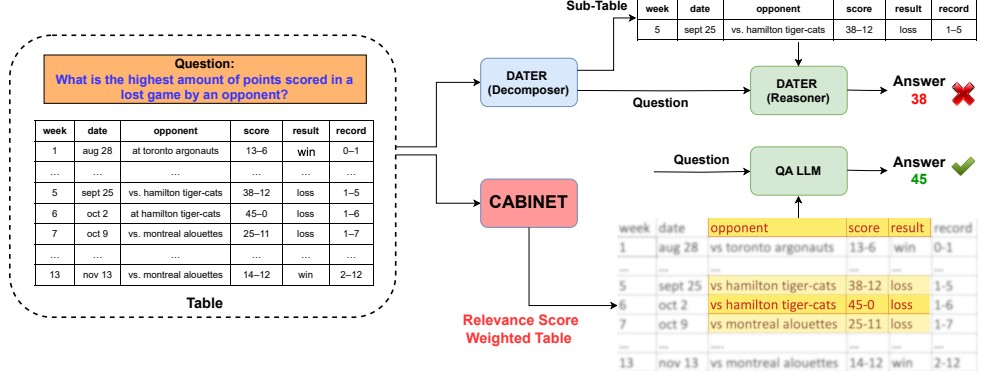

Figure 1: Comparison between CABINETand DATER (a GPT-3 based in-context learning method). For the given example, DATER extracts a wrong sub-table through hard decomposition (resulting in loss of useful information) that causes QA reasoner to answer incorrectly. CABINET weighs relevant table parts higher without removing content explicitly allowing QA LLM to answer correctly.

Significant efforts have been made to mitigate the issue of noise by pruning tabular data, albeit at cost of accuracy (Krichene et al., 2021), and by retrieving content from table for QA (Wang et al., 2022; Lei et al., 2023; Kumar et al., 2023). DATER (Ye et al., 2023), one of the state-of-the-art methods for table QA, proposed decomposing a table into simpler sub-tables containing information needed to answer the question by providing in-context examples to GPT-3 based Codex (Chen et al., 2021).

Such a question-conditioned hard decomposition of table is sub-optimal as the subsequent QA model cannot correct the error made during decomposition if relevant information is not selected (as shown in Figure 1). To mitigate this, we propose CABINET (**C**ontent Relev**A**nce-**B**ased No**I**se Reductio**N** for Tabl**E** Ques**T**ion-Answering) – a framework for table QA that weighs different table parts based on their relevance to the question without explicitly removing any content. CABINET comprises a relevance scorer (§ 3.1), which takes question and table as input to provide a relevance score to table content. The score is used to weigh corresponding content passed to the QA LLM, allowing it to focus more on the relevant content. The relevance scorer is unsupervised and trained with QA LLM differentiably due to lack of annotations denoting relevant table information. Although answer generation loss enables learning of relevance scorer, it acts as an indirect training signal.

Hence, to aid relevance scorer, inspired by how humans process tables, CABINET employs a parsing statement generator (§ 3.2) that describes which rows and columns are relevant to the question. For instance, consider the example in Figure 1, CABINET generates "consider rows with result as 'loss', and note the higher value in the 'score' column". The parsing statement is then used to identify corresponding cells, and their content is given more weight during relevance scoring. CABINET establishes new SoTA on three challenging table QA datasets (WikiTQ, FeTaQA and WikiSQL) significantly outperforming various strong baselines (§ 4.1). We show that CABINET is more robust to noise in tables and structural biases i.e. row and column ordering (§ 4.2). Further, the performance gains achieved by CABINET are even more pronounced for larger tables (§ 4.3), indicating that it successfully mitigates noisy table information irrelevant to a given question.

## 2 RELATED WORK

**Table Specific Architecture:** Tables contain information in a structured format, organized in rows and columns. Hence, many works have focused on developing table-specific models to utilize the semantics of table structure through its description. TabBERT (Yin et al., 2020) pre-trains BERT (Devlin et al., 2019) on paired table-text samples through masked language modeling (MLM). Deng et al. (2020) modified the bidirectional attention in BERT to incorporate table structure while performing MLM. TAPAS (Herzig et al., 2020) utilizes positional embeddings for rows and columns to explicitly capture cell location. Yang et al. (2022) noted that methods using positional embeddings are vulnerable to column and row permutations. To address this, they introduce TableFormer, a table-text encoding architecture that incorporates tabular structure through learnable attention biases. We show that LLMs become less susceptible to such permutations by learning to focus on relevant table parts through CABINET (§ 4.2).

**Table QA Specific Pre-training**: Eisenschlos et al. (2020) argued that the MLM objective to just fill in the blanks of table cells and descriptions is insufficient to capture relations between cells and associated text needed to perform table QA. They introduced additional pre-training tasks that require explicit question-table reasoning and complex table operations (such as aggregation). Other improvements include handling of numeric tokens (Han et al., 2022), temporal relations (Zhao et al., 2022), and selectively masking tokens that require table based reasoning (Gu et al., 2022). Methods like TAPEX (Liu et al., 2022), OmniTab (Jiang et al., 2022) etc. typically involve joint training over natural language-SQL pairs so that the underlying model learns to map the information implied in the question to the required table operations. However, as discussed in experiments (§ 4.2 and 4.3), these methods suffer significant performance drop when dealing with large and noisy tables owing to their limited capability at identifying information relevant to question.

**Few/Zero-Shot Learning with Large Language Models:** Given the remarkable performance of LLMs on various tasks without any task-specific training, their use for table understanding has also been explored extensively. Chen (2023) have shown that LLMs perform strongly on various table QA tasks using Chain of Thought (CoT) (Wei et al., 2022; Wang et al., 2023) prompting. Since typical LLMs are trained over unstructured text data, models specifically designed to handle structured data, such as StructGPT (Jiang et al., 2023) have also been used for table QA. LEVER (Ni et al., 2023) and BINDER (Cheng et al., 2023) utilized code-optimized GPT-Codex (Chen et al., 2021) to generate SQL statements that can be executed to answer questions over tabular data. DATER (Ye et al., 2023) uses Codex to break table into sub-tables conditioned on a given question through in-context learning. Such methods have no way to recover relevant table part to generate the correct answer in case it is omitted while generating sub-tables (as discussed in Figure 1).

## 3  METHODOLOGY

We summarize the architecture of CABINET in Figure 2. It comprises two components: **1) Unsupervised Relevance Scorer**, an unsupervised module comprising a transformer encoder that takes question and table as input and tokenizes them (steps 1 and 2 in Fig. 2) followed by assigning a relevance score to each table token (step 3 in Fig. 2). The relevance score is then multiplied with the corresponding token embedding at the time of giving it as input to QA LLM encoder (step 7 in Fig. 2). This ensures that noisy content with lower relevance score get suppressed and the QA LLM can focus on relevant tokens. The unsupervised relevance scorer is connected to QA LLM in a differentiable manner enabling it to be trained through answer generation loss (step 8 in Fig. 2).

Even though answer generation loss enables learning of unsupervised relevance scorer, it acts as an indirect training signal. To aid relevance scoring, we propose a weakly supervised module: **2) Relevant Cell Predictor through Table Parsing** that parses table conditioned on question to highlight cells containing relevant information (steps 4 and 5 in Fig. 2). It comprises two sub-steps where we first train a *Parsing Statement Generator* that describes criteria in natural language about which rows and columns should be used to derive the answer (step 4 in Fig. 2). Table cells corresponding to the criteria described in the parsing statement (step 5 in Fig. 2) are highlighted such that score for content tokens in highlighted cells is boosted by combining it with unsupervised relevance score through a linear combination (step 6 in Fig. 2). We conduct extensive ablations to establish efficacy for different modules (§ 4.4). We now discuss the details of each component.

### 3.1  UNSUPERVISED RELEVANCE SCORER

The unsupervised relevance scorer is used to assign a score to table content tokens. Since annotating cells of a table relevant to a given question is tedious, the relevance scorer is unsupervised and gets trained along with QA LLM through answer generation loss. Formally, consider a pair of table $\mathcal{T}$ and a question $\mathcal{Q}$ about $\mathcal{T}$. $\mathcal{Q}_{tokens} = \{q_1, q_2, ..., q_{|Q|}\}$ represents the question tokens, $\mathcal{T} = \{c_{ij} | 1 \leq i \leq \mathrm{N}_{row}, 1 \leq j \leq \mathrm{N}_{col}\}$, where $\mathrm{N}_{row}$ and $\mathrm{N}_{col}$ indicate number of rows and columns in $\mathcal{T}$ respectively, and $c_{ij}$ represents string in cell in the $i^{th}$ row and $j^{th}$ column. To make $\mathcal{T}$ suitable to be fed as input to a transformer-based LLM, we follow the commonly used linearising scheme (Liu et al., 2022) where table is flattened as (step 1 in Fig. 2):

$$\mathcal{T}_{flattened} = [HEAD]: c_{11} \,|\, c_{12} \,|\, \cdots \,|\, c_{1N_{col}} \,|\, [ROW]1: c_{21} \,|\, \cdots \,|\, c_{2N_{col}} \,|\, [ROW]2: \cdots \quad (1)$$

$[HEAD]$ and $[ROW]k$ indicate start of column header row and $k^{th}$ data row respectively. We separate special tokens and cell content using pipe symbol '|'. The string in Equation 1 is tokenized

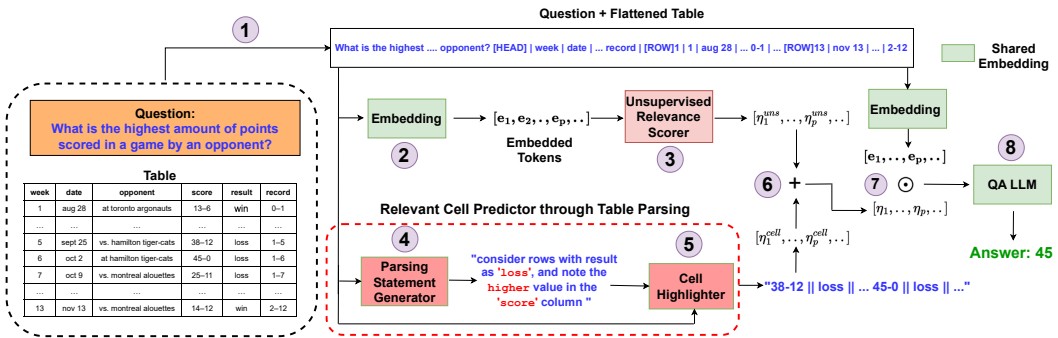

Figure 2: Overview of architecture of CABINET. The table is linearized (step 1) and embedded along with question through embedding layer of the underlying QA LLM (step 2). The embedded sequence is passed to the unsupervised relevance scorer that assigns a relevance score to each table token (step3). In parallel, the parsing statement generator describes the criteria for rows and columns relevant to deriving the answer (step 4) that is used to identify corresponding cells and assign a cell-based relevance score (step 5). The unsupervised and cell-based relevance is combined (step 6) and used to weigh the table content (step 7) to the QA LLM which generates the answer (step 8).

using the tokenizer of underlying QA LLM to obtain table tokens $\mathcal{T}_{tokens} = \{t_1, t_2, ..., t_{|\mathcal{T}_{tokens}|}\}$. $\mathcal{T}_{tokens}$ is concatenated to $Q_{tokens}$ to obtain $\mathcal{I}_{tokens} = (Q_{tokens}; \mathcal{T}_{tokens})$ which is given as input (steps 2 and 3 in Fig. 2) to Unsupervised Relevance Scorer (URS) comprising a transformer encoder $TE_{URS}$. The contextualized representation $h_p \in \mathcal{R}^d$ of the $p^{th}$ token is estimated as:

$$e_1^{URS}, e_2^{URS}, \cdots, e_{|\mathcal{I}_{tokens}|}^{URS} = Embedding_{URS}(\mathcal{I}_{tokens}) \tag{2}$$

$$h_1, \cdots, h_p, \cdots, h_{|\mathcal{I}_{tokens}|} = TE_{URS}(e_1^{URS}, e_2^{URS}, \cdots, e_{|\mathcal{I}_{tokens}|}^{URS}) \tag{3}$$

We aim to predict relevance score for each table token, however, since annotations for relevant table parts are unavailable, token relevance is not explicitly observable and we consider it as a latent variable. Further, we hypothesize that the representation space of table tokens can be structured better for modeling relevance by clustering their encodings into two categories - relevant and non-relevant. Variational Inference (VI) has been commonly used to estimate latent variable probability and group data points on the basis of latent topics (Srivastava & Sutton, 2017). Hence, we estimate relevance $\eta_p^{uns}$ of table token $t_p$ ($|Q_{tokens}| + 1 \leq p \leq |Q_{tokens}| + |\mathcal{T}_{tokens}|$) as (step 3 in Fig. 2):

$$\mu_p = \phi_\mu(h_p); \ \sigma_p = \phi_\sigma(h_p) \tag{4}$$

$$\eta_p^{uns} = sigmoid(z_p); \ z_p = \mu_p + s * \sigma_p \tag{5}$$

$s$ is sampled from standard normal distribution, $\phi_\mu$ and $\phi_\sigma$ are FC layers with weights $W_\mu \in \mathcal{R}^{d \times 1}$ and $W_\sigma \in \mathcal{R}^{d \times 1}$, $sigmoid$ is applied to normalize the relevance score in the range 0 to 1. To enable the relevance scorer to assign appropriate scores, we structure the latent space of $TE_{URS}$ by clustering table tokens into relevant and non-relevant. We use the method of van der Maaten & Hinton (2008) (details in appendix A.1) which performs clustering in a trainable manner using clustering loss $\mathcal{L}_{clu}$. We apply $\mathcal{L}_{clu}$ over latent representation $h_p$ of tokens which enables us to tune $TE_{URS}$ for clustering. During experiments, we observed that unit vectors for cluster centroids $\mu_{relevant}^{clu}$ and $\mu_{irrelevant}^{clu}$ are not well separated. To mitigate this, we enforce a separation loss $\mathcal{L}_{sep}$ that increases the distance between unit vectors representing cluster centroids:

$$\mathcal{L}_{sep} = 2 - \left|\left| \mu_{relevant}^{clu} - \mu_{irrelevant}^{clu} \right|\right|^2 \tag{6}$$

Further, it is desirable that relevance scores for tokens in one cluster (corresponding to irrelevant tokens) are low. To achieve this, we apply a sparsification loss $\mathcal{L}_{sparse}$ where the score logit $z_p$ is exponentiated with a negative coefficient to push logit values for relevant and irrelevant clusters to $\infty$ and $-\infty$ respectively that enables final score (after applying $sigmoid$) to be close to 1 and 0:

$$\mathcal{L}_{sparse} = \frac{1}{|\mathcal{T}_{tokens}|} \sum_p e^{-z_p^2}; \ |Q_{tokens}| + 1 \leq p \leq |Q_{tokens}| + |\mathcal{T}_{tokens}| \tag{7}$$

At the time of providing question and table as input to transformer encoder $TE_{QA}$ of the QA LLM, embedding ($e_p'$) corresponding to question tokens is used as is while embedding of each table token

is multiplied by its corresponding relevance score (steps 7 and 8 in Fig. 2):

$$e_1, e_2, \cdots, e_{|\mathcal{I}_{tokens}|} = Embedding_{QA}(\mathcal{I}_{tokens}) \tag{8}$$

$$e_p^{'} = \eta_p \odot e_p; \quad |\mathcal{Q}_{tokens}|+1 \leq p \leq |\mathcal{Q}_{tokens}| + |\mathcal{T}_{tokens}| \tag{9}$$

$$h_1^{'}, \cdots, h_{|\mathcal{I}_{tokens}|}^{'} = TE_{QA}(e_1^{'}, e_2^{'}, \cdots, e_{|\mathcal{I}_{tokens}|}^{'}) \tag{10}$$

$$a_1, a_2, \cdots, a_N = TD_{QA}(h_1^{'}, \cdots, h_{|\mathcal{I}_{tokens}|}^{'}) \tag{11}$$

'$\odot$' indicates scalar multiplication with vector operation, $TD_{QA}$ represents the transformer decoder of the QA LLM that generates the answer tokens $a_n$ sequentially. $TE_{URS}$, $TE_{QA}$ and $TD_{QA}$ are trained in an end-to-end manner through cross-entropy loss $\mathcal{L}_{CE}$ between the generated and ground-truth answer tokens. Thus, the total loss $\mathcal{L}$ becomes:

$$\mathcal{L} = \mathcal{L}_{CE} + \lambda_{clu} * \mathcal{L}_{clu} + \lambda_{sep} * \mathcal{L}_{sep} + \lambda_{sparse} * \mathcal{L}_{sparse} \tag{12}$$

The answer generation loss acts as an indirect training signal for relevance scorer. To aid unsupervised scorer, we propose a weakly-supervised module (trained separately from URS and QA LLM) that highlights relevant cells (discussed in next subsection). Table tokens for highlighted cells are assigned cell-based score $\eta_p^{cell}$ that is combined with unsupervised relevance score $\eta_p^{uns}$ through linear combination (step 6 in Fig. 2). Thus, final relevance score $\eta_p$ used in Eq. 9 is:

$$\eta_p = \lambda_{uns} * \eta_p^{uns} + \lambda_{cell} * \eta_p^{cell} \tag{13}$$

## 3.2 RELEVANT CELL PREDICTOR THROUGH TABLE PARSING

As discussed above, we train a separate module to highlight table cells relevant to a given question in a weakly-supervised manner. Since there is no Table QA dataset that contains annotations for table cells useful to answer a given question, we adopt a two-stage approach where we first train a *Parsing Statement Generator* (step 4 in Fig. 2) to generate a natural language text describing criteria for rows and columns relevant to the given question. Subsequently, we train another model that takes the parsing statement and table as input to identify the cells matching the criteria (step 5 in Fig. 2).

**Parsing Statement Generator (PSG)** comprises a pre-trained LLM - Flan T5-xl (Chung et al., 2022a) that is fine-tuned to take the question and table as input ($\mathcal{I}_{tokens}$) to generate a parsing statement $text_{parse}$ (step 4 in Fig. 2). The statement describes criteria stating which rows and columns are useful to derive the answer. To bootstrap training of PSG, we manually annotate very few ($\sim$300) question-table pairs with parsing statement. For instance, for table and question shown in Figure 2, we annotate parsing statement as '*To derive answer, note the values of higher score in rows with result as loss*'. To circumvent annotating samples for each table QA dataset, we choose WikiTableQuestions (WikiTQ) dataset (Pasupat & Liang, 2015) to select the samples for annotation since it is the most complex QA dataset containing a variety of samples. We sample diverse set of questions for annotation (please refer appendix A.2 for details and examples). The sampled question along with its table are manually annotated with parsing statement which is used to fine-tune PSG. The trained PSG model is then used to generate parsing statement for any question-table pair from datasets studied for experiments. We release the dataset of manually annotated parsing statements.

**Cell Highlighting based on Parsing Statement:** To identify table cells for the criteria described in the parsing statement $text_{parse}$, we need a way to map the statement to corresponding cells. To this end, we use ToTTo dataset (Parikh et al., 2020) that contains samples of (table, list of highlighted cell coordinates) pairs. Each pair is accompanied by a text description summarising the content of the corresponding list of cells. We fine-tune a cell highlighting model $Cell\_Highlighter_{LLM}$ comprising of Flan T5-xl on ToTTo dataset where the table and summarising text are given as input to generate the content of corresponding highlighted cells. Once $Cell\_Highlighter_{LLM}$ is trained, we provide the table and $text_{parse}$ as input to identify and generate content of corresponding cells. For instance, consider example in Figure 2, given the parsing statement shown in figure as input, the cell predictor generates $'38 - 12 \parallel loss \parallel 45 - 0 \parallel loss \parallel...'$ (step 5). More formally,

$$c_1^{highlighted} \parallel \cdots \parallel c_M^{highlighted} = Cell\_Highlighter_{LLM}(\mathcal{T}, text_{parse}) \tag{14}$$

$c_r^{highlighted}$ represents the string of $r^{th}$ highlighted cell predicted based on parsing statement. $M$ is a variable number, '$\parallel$' is a delimiter to separate cell content. For $1 \leq r \leq M$, if $c_r^{highlighted}$ exactly matches with the content of some cell in $\mathcal{T}$, then the tokens $t_p$ of matching cell is assigned a cell relevance score ($\eta_p^{cell}$) of 1. $\eta_p^{cell}$ is set to 0 for table tokens belonging to cells in $\mathcal{T}$ whose content does not match with any $c_r^{highlighted}$. $\eta_p^{cell}$ is then combined with unsupervised relevance score $\eta_p^{uns}$ as in Eq. 13. We now discuss experiments performed to validate the efficacy of our approach.

## 4 EXPERIMENTS AND EVALUATION

**Implementation Details:** For the encoder ($TE_{QA}$) and decoder ($TD_{QA}$) of the QA LLM, we employ the OmniTab (Jiang et al., 2022) backbone (pre-trained for table understanding) comprising of BART-Large (Lewis et al., 2020). The embeddings of unsupervised relevance scorer (URS), $Embedding_{URS}$, and QA model, $Embedding_{QA}$, are shared. URS encoder ($TE_{URS}$) is initialized with the architecture and weights of QA LLM encoder ($TE_{QA}$), though they do not share weights during QA training. Consequently, the hidden dimension $d$ of $TE_{URS}$ is 1024. We train CABINET and baselines (wherever needed) for 30 epochs on an effective batch size (BS) of 128 using 8 80GB A100 GPUs (BS of 8/GPU with gradient accumulation 2) using a learning rate of $1e-5$ with cosine annealing (Loshchilov & Hutter, 2017) through AdamW optimizer (Loshchilov & Hutter, 2019).

**Datasets and Evaluation Metrics:** We evaluate CABINET on three commonly used datasets – *(i)* **WikiTableQuestion (WikiTQ)** (Pasupat & Liang, 2015) which is one of the most commonly used and highly complex datasets consisting of about 2100 HTML tables from Wikipedia and about $22,033$ questions that require complex operations such as comparison, aggregation, and arithmetic operations to arrive at the answer; *(ii)* **FeTaQA** (Nan et al., 2022), a challenging dataset consisting of about $10,000$ questions that have a long-form natural language answer (18 words on average) such that it requires fetching multiple entities from the table, aggregating and reasoning over these entities, and structuring the inferred information to produce a coherent answer; and *(iii)* **WikiSQL** (Zhong et al., 2017) that comprises roughly $80,654$ questions over $24,241$ Wikipedia tables. It also provides the equivalent SQL query for each question to obtain the correct answer. While we do not generate SQL (or other implicit logical forms) and only use the natural language questions and answers from this dataset, it serves as a useful benchmark to compare CABINET with table understanding methods that generate explicit logical forms to extract relevant answers from table. The ground-truth answers in both WikiTQ and WikiSQL datasets are short (1-2 words). Hence, we use exact-match accuracy (Acc.) to compare various methods. For FetaQA dataset, ground-truth answers being long-form ($\approx 18$ words on average), we employ commonly used overlap-based metric Sacre-BLEU (S-BLEU) (Post, 2018). We report performance on test split for all datasets.

### 4.1 PERFORMANCE OF CABINET ON TABLE QA

We present a detailed comparative analysis of results achieved by CABINET with a variety of baselines. We consider three different categories of methods – *(i)* LLMs specifically pre-trained for table understanding and fine-tuned for QA, such as TAPEX (Liu et al., 2022), ReasTAP (Zhao et al., 2022) and OmniTab (Jiang et al., 2022); *(ii)* fine-tuning LLMs (pre-trained on text only) such as T5-3b (Raffel et al., 2020) and Flan T5-xl (Chung et al., 2022b); and *(iii)* few or zero shot prompting of LLMs like Struct-GPT (Jiang et al., 2023) and approaches that employ such LLMs for in-context learning like LEVER (Ni et al., 2023), BINDER (Cheng et al., 2023) and DATER (Ye et al., 2023).

Table 1 presents the performance of various methods on the WikiTQ dataset, and we can observe CAB-INET with an accuracy of **69.1**% outperforms the best-performing baselines in each of the three categories and establishes new state-of-the-art. Specifically, CABINET outperforms OmniTab, DATER, and fine-tuned Flan T5-xl by 6.4%, 3.2% and 4.7%, in absolute terms, respectively. Also, note that simple prompting of ChatGPT does not work well for Table QA. We want to highlight that CABINET performs much better than GPT-3 and Codex-based SoTA in-context learning methods despite containing orders of magnitude fewer parameters.

Table 1: Comparison of CABINET with different baselines on WikiTQ. CABINET achieves significantly better accuracy.

| Method | Acc. | # params |
|---|---|---|
| **Fine-tuning Table-specific LLMs** | | |
| TAPAS (Herzig et al., 2020) | 48.8 | 345 M |
| TaBERT (Yin et al., 2020) | 52.3 | 345 M |
| MATE (Eisenschlos et al., 2021) | 51.5 | 340 M |
| GraPPa (Yu et al., 2021) | 52.7 | 355 M |
| DoT (Krichene et al., 2021) | 54.0 | 299 M |
| TableFormer (Yang et al., 2022) | 52.6 | 345 M |
| TAPEX (Liu et al., 2022) | 55.5 | 405 M |
| ReasTAP (Zhao et al., 2022) | 58.6 | 406 M |
| TaCube (Zhou et al., 2022) | 60.8 | 406 M |
| OmniTab (Jiang et al., 2022) | 62.7 | 406 M |
| | | |
| **Fine-tuning text-based LLMs** | | |
| T5-3b (Xie et al., 2022)) | 49.3 | 2.9 B |
| FlanT5-xl (Chung et al., 2022a) | 64.4 | 2.9 B |
| | | |
| **Few/zero shot Prompting of LLMs** | | |
| Codex (Ye et al., 2023) | 47.6 | 175 B |
| Codex-COT (Chen, 2023) | 48.8 | 175 B |
| Binder (Cheng et al., 2023) | 64.6 | 175 B |
| LEVER (Ni et al., 2023) | 65.8 | 175 B |
| DATER (Ye et al., 2023) | 65.9 | 175 B |
| ChatGPT (Jiang et al., 2023) | 43.3 | 175 B |
| StructGPT (Jiang et al., 2023) | 48.4 | 175 B |
| | | |
| **CABINET (Ours)** | **69.1** | 560 M |

Table 2: Comparison with different categories of baselines on FeTaQA. CABINET achieves significantly better Sacre-BLEU (S-BLEU).

| Method | S-BLEU | # params |
|---|---|---|
| **Fine-tuning Table-specific LLMs** | | |
| PeaQA (Pal et al., 2022) | 33.5 | 406 M |
| TAPEX (Liu et al., 2022) | 34.7 | 406 M |
| OmniTab (Jiang et al., 2022) | 34.9 | 406 M |
| **Fine-tuning text-based LLMs** | | |
| T5-small (Nan et al., 2022) | 21.6 | 60 M |
| T5-base (Nan et al., 2022) | 28.1 | 222 M |
| T5-large (Nan et al., 2022) | 30.5 | 738 M |
| T5-3b (Xie et al., 2022) | 33.4 | 2.9 B |
| FlanT5-xl | 36.2 | 2.9 B |
| **Few/zero shot Prompting of LLMs** | | |
| Codex-COT (Chen, 2023) | 27.0 | 175 B |
| Codex (Ye et al., 2023) | 27.9 | 175 B |
| DATER (Ye et al., 2023) | 30.9 | 175 B |
| **CABINET (Ours)** | **40.5** | 560 M |

Table 3: Comparison with different categories of baselines on WikiSQL. CABINET achieves better Accuracy (Acc.).

| Method | Acc. | # params |
|---|---|---|
| **Fine-tuning Table-specific LLMs** | | |
| TAPAS (Herzig et al., 2020) | 86.4 | 345 M |
| GraPPa (Yu et al., 2021) | 84.7 | 355 M |
| DoT (Krichene et al., 2021) | 85.5 | 299 M |
| TAPEX (Liu et al., 2022) | 86.4 | 406 M |
| OmniTab (Jiang et al., 2022) | 87.9 | 406 M |
| UTP (Chen et al., 2023b) | 88.1 | 345 M |
| ReasTAP (Zhao et al., 2022) | 88.8 | 406 M |
| **Fine-tuning text-based LLMs** | | |
| T5-3b (Xie et al., 2022) | 85.9 | 2.9 B |
| FlanT5-xl | 87.8 | 2.9 B |
| **Few/zero shot Prompting of LLMs** | | |
| ChatGPT (Jiang et al., 2023) | 51.6 | 175 B |
| StructGPT (Jiang et al., 2023) | 54.4 | 175 B |
| **CABINET (Ours)** | **89.5** | 560 M |

Similar observations hold for FeTaQA (Table 2) and WikiSQL (Table 3) datasets where CABINET achieves new SoTA performance. For generating long descriptive answers for questions in FeTaQA, CABINET achieves SoTA S-BLEU of $40.5$ outperforming OmniTab, fine-tuned Flan T5-xl and DATER by a margin of $5.6$, $4.3$ and $9.6$ absolute percentage points, respectively. We report performance only for baselines that have explored the dataset in their work (except for T5 and Flan T5). We use their code/API for evaluation if available or else specify performance as reported in their paper. Similarly, for WikiSQL dataset, CABINET pushes the SoTA by $0.7\%$ on already high performance of ReasTAP (current SoTA). Since best performance on WikiSQL is already high, the absolute performance gains of $0.7\%$ should be interpreted as a proportion of scope of further improvement possible, i.e., $0.7/(100 - 88.8)$, which is $\approx 6\%$.

## 4.2 How Robust is CABINET to Noise and Irrelevant Information?

Despite the remarkable success of transformer-based models on table understanding, they are sensitive to noise and perturbations to the tabular data (Pi et al., 2022; Yang et al., 2022; Zhao et al., 2023). We examine the robustness and sensitivity of CABINET towards noise while performing Table QA. We introduce noise by perturbing tables in test split and report the relative percentage drop in performance. We perform four types of perturbations: **1) Row Addition (RA):** insert noise into a table by adding rows from another table that contains same number of columns; **2) Row Permutation (RP):** randomly permute ordering of rows (Pi et al., 2022); **3) Column Permutation (CP):** randomly permute column ordering; and **(4) Cell Replacement (CR):** replace content of certain cells with content from some other table. We perform each perturbation separately to obtain four perturbed test splits for each dataset. Please see appendix A.5 for further details about the procedure.

Figure 3 summarizes the relative drop in performance of CABINET and the dataset-specific best baseline for the three datasets. Note that for all the perturbation categories, CABINET leads to significantly less drop in performance when compared with the corresponding baseline, highlighting the robustness and ability of CABINET to identify the relevant portions of the underlying table. Specifically, CABINET is significantly less sensitive to row and column permutations (RP and CP), indicating that relevance scoring of tokens helps the QA LLM to focus more on relevant information and reduces the potential ordering biases commonly observed in models pre-trained on tabular data (Yang et al., 2022). For the cell replacement (CR) and row addition (RA) perturbations, where extraneous information is explicitly added to the table, the drop in performance suffered by CABINET is significantly less compared to the baselines owing to the superior ability of CABINET to identify relevant information. For instance, in the case of WikiTQ, the relative drop in performance for RA is $\approx 19\%$ for OmniTab, almost $40\%$ higher than CABINET ($\approx 11.5\%$). This consistent trend holds for FeTaQA and WikiSQL datasets as well.

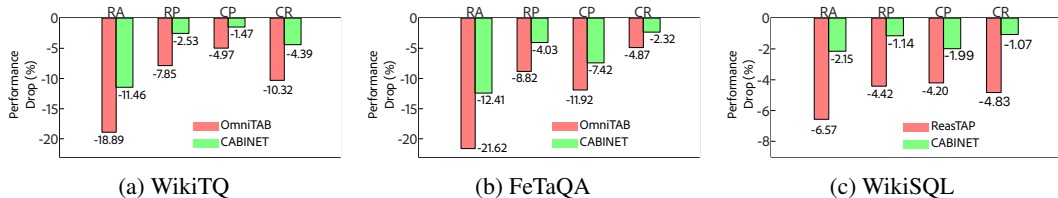

Figure 3: Relative performance drop (%) with perturbations (RA - Row Addition, RP - Row Permutation, CP - Column Permutation, CR - Cell Replacement). We compare CABINET (green) with OmniTab (red) on WikiTQ and FeTaQA ; and against ReasTAP (red) on WikiSQL. CABINET is more robust to addition of noise to table and shuffling of row and column ordering.

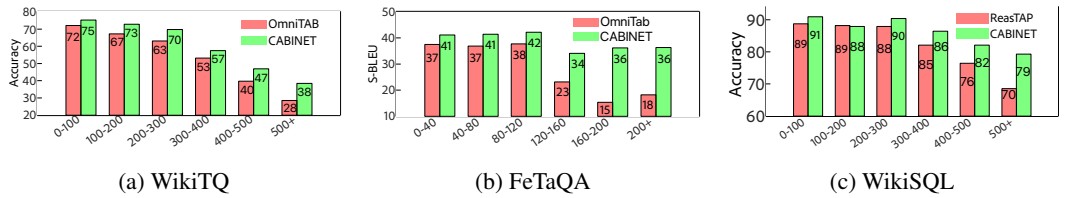

Figure 4: Variation in performance with table size (# cells). We compare CABINET (green) with OmniTab (red) on WikiTQ (left) and FeTaQA (middle), and against ReasTAP (red) for WikiSQL (right). It can be seen that CABINET performs much better than the baselines on larger tables.

### 4.3  IMPACT OF TABLE SIZE ON PERFORMANCE

We now study how CABINET performs with tables of different sizes. Tables typically comprise a large amount of data, so the entire information is usually not required to answer a given question and acts as *distracting information* (Neeraja et al., 2021). This noise or irrelevant data poses a severe challenge for table understanding models and leads to poor generalization for larger tables (Kumar et al., 2023; Chen, 2023). We consider the number of cells in the table as a proxy for its size and bin all the questions in the three datasets into six categories based on the number of cells (Figure 4) and compare the performance of CABINET with dataset-specific best-performing baseline. We note that for all the datasets, while model performance drops with increasing table size, CABINET consistently and significantly outperforms the baseline methods across all table size categories. Moreover, the differences become starker for larger tables. For instance, for the largest tables in FeTaQA, CABINET achieves double the S-BLEU scores compared to OmniTab (36 vs. 18). Similarly, for the other two datasets, CABINET achieves significantly high performance for the largest tables ($> 500$ cells) compared to the baselines – accuracy of 38 vs. OmniTab's 28 for WikiTQ and 79 vs. ReasTAP's 70 for WikiSQL. These empirical observations provide further evidence for CABINET's ability to identify relevant content, making the QA LLM relatively robust to table size.

### 4.4  DISCUSSION ON THE IMPACT OF DIFFERENT DESIGN CHOICES FOR CABINET

**Effect of Clustering Table Tokens:** We study the impact of clustering the table tokens using their latent representations (discussed in Section 3.1). To do so, we toggle the clustering loss ($\mathcal{L}_{clu}$), cluster centroids separation loss ($\mathcal{L}_{sep}$), and score sparsification loss ($\mathcal{L}_{sparse}$) by setting their weight ($\lambda_{clu}, \lambda_{sep}, \lambda_{sparse}$) to 0 or 1. For this study, we only use unsupervised relevance scorer by turning off weakly supervised cell predictor to eliminate other influencing factors. Results are summarized in Table 4 where we can observe that applying all three losses yields the best performance (row 6). Specifically, for WikiSQL, clustering improves performance when score sparsification loss is applied (row 4 vs. row 3) which is due to the fact that sparsification enables categorizing scores into low and high. For WikiTQ and FeTaQA, adding the cluster centroids separation loss further increases the efficacy of clustering and sparsification yielding the best results.

**Combining Unsupervised Relevance Scorer with Cell Predictor:** We vary the relative importance given to relevance score predicted by unsupervised relevance scorer and weakly-supervised cell predictor by varying $\lambda_{uns}$ and $\lambda_{cell}$ in Eq. 13. Table 5 shows that combining the two modules yields much better accuracy for WikiTQ and FeTaQA compared to just using unsupervised relevance scorer (row 1 vs. row 2). This highlights that the weakly-supervised cell predictor complements unsuper-

Table 4: Effect of applying clustering ($\mathcal{L}_{clu}$), centroid separation ($\mathcal{L}_{sep}$) and relevance score sparsification loss ($\mathcal{L}_{sparse}$). Clustering table tokens by enforcing sparsity in relevance scores and distance between cluster centroids improves performance.

| $\mathcal{L}_{clu}$ | $\mathcal{L}_{sep}$ | $\mathcal{L}_{sparse}$ | WikiTQ | FeTaQA | WikiSQL |
|---|---|---|---|---|---|
| ✗ | ✗ | ✗ | 60.8 | 35.1 | 86.2 |
| ✗ | ✗ | ✓ | 60.9 | 35.1 | 86.3 |
| ✓ | ✗ | ✗ | 62.7 | 35.0 | 88.9 |
| ✓ | ✗ | ✓ | 61.0 | 35.0 | **89.5** |
| ✓ | ✓ | ✗ | 61.0 | 35.1 | 89.1 |
| ✓ | ✓ | ✓ | **65.6** | **35.8** | 89.3 |

Table 5: Impact of combining unsupervised relevance score (weight $\lambda_{uns}$) and weakly-supervised cell-based relevance score (weight $\lambda_{cell}$). Fusing the relevance from both components gives optimal performance.

| $\lambda_{uns}$ | $\lambda_{cell}$ | WikiTQ | FeTaQA, | WikiSQL |
|---|---|---|---|---|
| 1 | 0 | 65.6 | 35.8 | **89.2** |
| 0.7 | 0.3 | **69.1** | **40.5** | **89.2** |
| 0.5 | 0.5 | 68.6 | **40.5** | 88.9 |
| 0.3 | 0.7 | 67.0 | 38.9 | 88.8 |
| 0 | 1 | 37.6 | 24.2 | 34.1 |

Figure 5: Visualisation depicting that Unsupervised Relevance Scorer (URS) assigns higher score to table parts relevant to the question (rows where "two and a half men" either won or got nominated for an award). Further, the weakly-supervised parsing statement based relevant cell predictor identifies the cells for the row missed by URS (year 2006, golden icon award best actor - comedy series)

vised scorer by identifying further relevant table content (Figure 5 depicts qualitative visualisation for the same). For WikiSQL, same performance is observed with and without the cell predictor. Further it is observed that using only the cell predictor (last row) achieves significantly low performance due to the fact that the number of cells highlighted by the cell predictor is much lesser resulting in assigning a score of zero to most table content in cases where it misses to identify important cells.

We show CABINET can be used with TAPEX backbone (instead of OmniTab) to improve it's performance showing generality of our framework (Appendix A.7). We show that giving parsing statement as input to QA LLM, replacing URS with BERT based similarity metric for relevance scoring, and using question directly instead of parsing statement to generate highlighted cells gives sub-optimal performance compared to CABINET, justifying our design choice (Appendix A.8). Appendix A.10 shows case study depicting how clustering losses interact to yield improvements. Appendix A.11 shows that CABINET can be used to improve other NLP tasks like reading comprehension.

## 5 CONCLUSIONS

We studied the problem of question-answering over tables and focused on identifying the relevant portions of the table to derive the answer. Generally, only a small subset of the tabular data is required to answer the question, and owing to the vulnerability of LLMs to noise, the extraneous information leads to sub-optimal performance. This problem is further exacerbated in the case of large tables. Our proposed framework, CABINET addresses this issue by weighing the table content based on its relevance to the question, identifying the relevant rows and columns, and highlighting the content of the relevant cells. CABINET establishes new SoTA on three commonly used challenging benchmarks, outperforming table-specific models, as well as methods that employ in-context learning with much larger GPT-3 scale models. We show empirically that CABINET is more robust to noise and generalizes well for larger tables, indicating its efficacy in mitigating noise and overcoming table structural biases typically learned during training.

## 6 ETHICS AND REPRODUCIBILITY STATEMENT

We use publicly available datasets and LLMs (which are commonly used) to conduct the study in our work. The only data that we annotate is $\sim 300$ samples of table-question pairs with parsing statement describing rows and columns relevant to question. The parsing statement were written keeping in mind the safety and ethics guidelines without any potential concerns. To encourage reproducibility, we release our code and datasets (including manually written parsing statements) at this link. We describe the details of the datasets in § 4 (under 'Datasets and Evaluation Metrics') and the LLMs used in § 4 (under 'Implementation Details') and § 3.2. Further, we provide the implementation details of our method in § 4 (under 'Implementation Details') and discuss baselines used for comparison in § 4.1. Finally, we elaborate further details of our method in Appendix - Trainable clustering over latent representation of table tokens (A.1), Details of parsing statement annotation procedure (A.2) and Further details on table perturbation procedure (A.5).

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

# A    APPENDIX

## A.1    CLUSTERING LATENT VECTORS

As discussed in Section 3.1, the table tokens are clustered in a trainable manner (van der Maaten & Hinton, 2008) using their latent representations encoded through Unsupervised Relevance Scorer (URS). We discuss the details of the trainable clustering algorithm.

Formally, the probability of latent vector $h_p$ corresponding to the $p^{th}$ token belonging to $j^{th}$ cluster is given by Equation 15

$$q_{pj} = \frac{(1 + ||h_p - \mu_j^{clu}||^2/\alpha)^{-\frac{\alpha+1}{2}}}{\sum_{j'}(1 + ||h_p - \mu_{j'}^{clu}||^2/\alpha)^{-\frac{\alpha+1}{2}}} \tag{15}$$

Here, $h_p$ is the contextualised latent vector of the $p^{th}$ token obtained using $TE_{URS}$, $\alpha$ is the degrees of freedom of the Student's t distribution, $\mu_j^{clu}$ is the centroid of the $j^{th}$ cluster and $j \in \{0, 1\}$. Here, $\mu_0^{clu} = \mu_{relevant}^{clu}$ and $\mu_1^{clu} = \mu_{irrelevant}^{clu}$. Moreover, the cluster centroids are learnable.

The clustering process is iteratively refined by enforcing KL divergence minimization between the probability distribution for each token and a pseudo distribution generated using $q_{pj}$. Mathematically, the clustering loss is

$$\mathcal{L}_{clu} = \frac{1}{B}\sum_b KL(Z||Q) = \frac{1}{B}\sum_b\sum_p\sum_j z_{pj}log\frac{z_{pj}}{q_{pj}} \tag{16}$$

where $b \in \{1, 2, \cdots, B\}$, $B$ is the batch size, and $Z$ is the target distribution. A naive approach to model $Z$ would be setting each $z_p$ to a delta distribution (to the nearest centroid) for representations above a confidence threshold and ignoring the rest. However, because $q_p$ are soft assignments, it is more natural and flexible to use softer probabilistic targets. So, we model $z_{pj}$ with Equation 17

$$z_{pj} = \frac{q_{pj}^2/f_{pj}}{\sum_{j'} q_{pj'}^2/f_{pj'}}, \text{ where } f_{pj} = \sum_p q_{pj} \text{ is the soft cluster frequency} \tag{17}$$

## A.2    DETAILS OF ANNOTATING TABLE PARSING STATEMENT

As discussed in Section 3.2, we train a parsing statement generator that generates a criteria describing which rows and columns contain information relevant to the table. The generated parsing statement is used to identify corresponding cells which are then combined with the relevant table part detected by the unsupervised relevance scorer through relevance scoring. To bootstrap the training of the parsing statement generator, we manually annotate ∼300 question-table pairs from the WikiTQ dataset with parsing statement. To select samples to be annotated, we sample questions from WikiTQ as it consists of a diverse and complex set of reasoning questions over tables, thereby sampling questions from this dataset allows us to select a set that is representative of questions in other datasets for Table QA.

To select a diverse set of questions from WikiTQ for annotation, we group questions into 5 clusters by clustering their representations such that we randomly sample an equal number of questions from each cluster. Specifically, we use an encoder-only model DeBERTa-V2 to encode the questions. Since the questions pertain to tables, we embed questions in train split of WikiTQ through an LLM possessing table understanding - DeBERTaV2 (He et al., 2023) initialized with weights tuned through table understanding task proposed by PASTA (Gu et al., 2022).

Once the questions are clustered, we sample a small subset (2.5%) from each cluster to manually annotate with the parsing statement. Some examples of questions, the parsing statement and the corresponding answer from each cluster can be seen in Table 6.

Table 6: Examples of parsing statement annotated manually for questions sampled from each cluster.

| Cluster | Question | Answer | Parsing Statement |
|---|---|---|---|
| 1 | how many episodes had a nightly rank of 11? | 3 | to find number of episodes with nightly rank of 11, we need to look at the column named "nightly rank" and count number of times the value 11 occurs. |
| | how many games during this season were aired on cbs? | 3 | to find number of games aired on cbs, we need to look at the column tv and retrieve the rows having value cbs. from table, there are 3 occurences of cbs in tv column. |
| 2 | which season was more successful, 1995/96 or 1996/97? | 1996/97 | to compare between success of the seasons 1995/96 or 1996/97, we need to look at the final place in both the seasons. from table, place corresponding to season 1995/96 is 19th, which is bad compared to the place 1st corresponding to season 1996/1997. |
| | did john howard serve as prime minister for more or less time than julia gillard? | more | for answering this question, we need to look at total time in office for both john howard and julia gillard. for prime minister john howard total time in office is 4,284 days which more than 1,099 days i.e., total time in office for julia gillard. |
| 3 | which party won the top place in the election? | Australian Labor Party | to find party with top place in the election, we need to compare the seats of each party. from table, australian labor party has the maximum number of seats. |
| | which role is the most common from all the titles? | Salesman | most common title refers to title which occurs the most number of times. from table, in the column role, the value salesman occurs the most number of times. |
| 4 | what publication is listed before play magazine? | Nintendo Power | from table, in the column publication, the row before play magazine has the value nintendo power |
| | what college has the top enrollment? | Cornell University | to find the top enrollment, we need to find college with maximum number of enrollment. from table, the maximum value of enrollment column is 20,400 corresponding to the institution cornell university. |
| 5 | what is the difference between the caps of henry carlsson and borge leander? | 1 | to find difference of caps between henry carlson and borge leander, we need to first obtain the caps values. from table, caps of name henry carlson i.e., henry "garvis" carlson is 5 and caps of borje leander is 4. so difference is 1. |
| | what was the difference in time between the 8th place finisher and the first place finisher? | +17.32 | to find difference between 8th place finisher and first place finisher, we need to look at the difference column corresponding to rank 8 in the table. from table, the value is +17.32. |

### A.3 Additional Related work

Graphs have been used to capture table structure and similarity between table-question samples across the dataset (Iyer et al., 2023). TANDA (Garg et al., 2020) performs the task of answer sentence selection (AS2) for a given question by fine-tuning the transformer-based model to select the right candidate from answer candidates. They first fine-tune an LLM on the AS2 task followed by adapting it to a specific domain. Zhang et al. (2021) built over TANDA by considering remaining answer candidates as evidence while deciding the appropriateness of a particular answer candidate. This was further extended by Iyer et al. (2023) who modeled the relation between other similar question samples and corresponding answers in the dataset with the given question and answer candidates using a graph. In our current work, we mainly focussed on identifying relevant parts of the table useful to derive the answer to the question, however, a similar method can also be explored as future work that utilizes a graph to capture the table structure (positioning of different cells in the table) for a given sample and also model the similarity between multiple question-table pairs across the dataset. Another direction focuses on applying semantic parsing over the input text (question) and table to generate a logical form (such as SQL) which when executed fetches relevant information (Yu et al., 2021).

### A.4 Performance on Numeric Vs Non-numeric & Retrieval Vs Aggregation-based Answers

We analyse the performance of CABINET on generating answers of distinct types by categorizing them into four categories: *numeric*, *non-numeric*, *retrieval*, and *non-retrieval (aggregation)*. This meticulous categorization allows us to gain a nuanced understanding of how CABINET performs against baselines for table-question pair that require different types of answers to be generated.

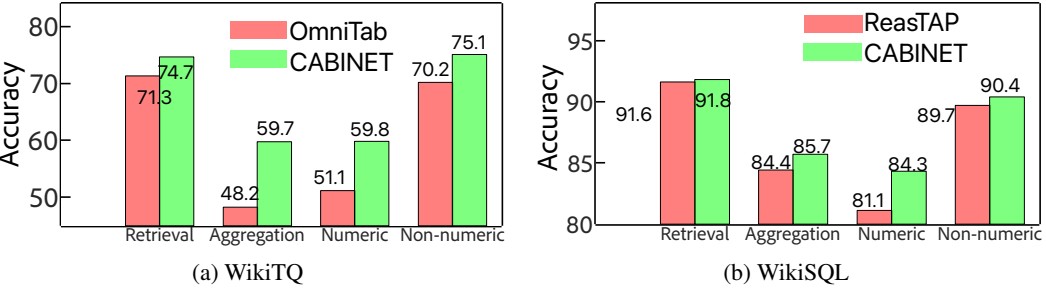

Figure 6: Performance comparison between CABINET with OmniTab on WikiTQ and with ReasTAP on WikiSQL when test samples are categorised based on the answer type - retrieval, non-retrieval (aggregation), numeric and non-numeric. We see CABINET provides significantly better performance on all answer type categories. It's noteworthy to mention that this analysis excludes FeTaQA, as it comprises of free-form long answers, hence categorization is not possible.

Figure 6 summarises the results. For both WikiTQ and WikiSQL, our method consistently demonstrates better performance for all four categories. We see an improvement of around $8 - 10\%$ for WikiTQ and around $3 - 5\%$ for WikiSQL for numeric and non-retrieval type answers, thereby highlighting the improved aggregation capabilities using CABINET.

### A.5 Details of Table Perturbation

In Section 4.2, we discussed the impact of perturbing the tables on performance. We discuss the detailed steps that we followed to inject the 4 types of perturbation separately in the tables. **For Row Addition**, we create sets of table with same number of columns. Further, based on the number of cells $m$ in a table, we insert $n$ rows from another table with the same number of columns at a random position. The exact scheme followed is, 1) $n = 1$ if $m \leq 150$, 2) $n = 2$ if $m > 150$ & $m \leq 300$, 3) $n = 5$ if $m > 300$ & $m \leq 450$ and 4) $n = 8$ if $m > 450$. For **row permutation** and **column permnutation**, we randomly permute the rows and columns respectively without any specific conditioning over the number of rows and columns present in the table. For

**cell replacement**, we divide the dataset again into the 4 buckets same as that of row addition and randomly replace $0.02\%$, $0.05\%$, $0.1\%$ and $0.12\%$ of the total number of cells in the table.

## A.6 ADDITIONAL IMPLEMENTATION DETAILS

As stated in section 4 (under implementation details) of the main paper, we employ the OmniTab backbone built over BART-Large architecture for the encoder ($TE_{QA}$) and decoder ($TD_{QA}$) of the QA LLM. Further, URS encoder ($TE_{URS}$) is initialized with the architecture and weights of QA LLM encoder ($TE_{QA}$), though, they do not share weights during training. However, the embedding layers $Embedding_{URS}$ and $Embedding_{QA}$ share weights. The hidden dimension $d$ of $TE_{URS}$ is 1024, similar to that of both $TE_{QA}$ and $TD_{QA}$. We train CABINET and other baselines for 30 epochs on an effective batch size (BS) of 128 using 8 80 GB A100 GPUs (BS of 8/GPU with gradient accumulation 2) using learning rate of $1e^{-5}$ with Cosine Annealing through AdamW optimizer. We carry out hyper-parameter tuning based on the validation set to come up with the optimal values of learning rate ($1e^{-5}$), scheduler (Cosine Annealing), batch size (8), gradient accumulation steps (2) and the optimizer (AdamW). We leverage text pre-trained Flan T5-xl as the backbone for Parsing Statement Generator (PSG) and the Cell Highlighter LLM. The Parsing Statement Generator is trained for 50 epochs on the 300 manually annotated question-table-parsing statement triplets with hyper-parameters setting same as that of CABINET, however the only difference being the effective batch size of 16 (BS of 2/GPU with 8 GPUs without gradient accumulation). On a similar note, the Cell Highlighter LLM is trained for 3 epochs on the ToTTo dataset to generate highlighted cells given the table and a natural language statement describing a subpart of the table as the input. During the inference phase of generating answer, parsing statements and highlighted cells, we use beam search decoding with a beam size of 3.

## A.7 CAN CABINET BE USED WITH OTHER QA LLM BACKBONES TO IMPROVE PERFORMANCE?

To verify the generality of CABINET as a framework that can be used with any encoder-decoder style Table QA method, we use TAPEX (Liu et al., 2022) as the backbone to initialise the unsupervised contextual relevance scorer (instead of OmniTab). TAPEX is also used as the underlying QA LLM. Table 7 summarises the results where it can be seen that the performance improves on all the three datasets. This highlights the generality of CABINET in improving performance.

Table 7: Performance improvement achieved through CABINET over TAPEX by using TAPEX as the underlying QA LLM (instead of OmniTab). Further, the encoder of TAPEX is used to initialise the unsupervised relevance scorer (instead of OmniTab). This highlights the generality of CABINET as a framework for improving performance.

| Method | WikiTQ | FeTaQA | WikiSQL |
|---|---|---|---|
| TAPEX | 55.5 | 34.6 | 86.4 |
| **CABINET w TAPEX Backbone** | **62.7** | **37.8** | **87.3** |

## A.8 ABLATIONS ON ALTERNATE DESIGN CHOICES FOR CABINET

We discuss several explorations on alternate ways of utlising the different components of CABINET framework. The results pertaining to those can be seen in table 8.

1. Instead of using the generated parsing statement to highlight cells for our cell-based scoring mechanism, we feed the parsing statement in the input to the QA LLM along with the question and table in order to verify if parsing statement alone can improve the QA performance when directly used as an instruction to the model. We use the unsupervised relevance scorer along with the QA LLM. Table 8 summarises the results. It can be seen that using the parsing statement as input to the QA model gives significantly sub-optimal performance (row 2 vs row 7 in Table 8) compared to highlighting cells to obtain the corresponding scores that can be combined with the relevance scores predicted by the unsupervised relevance scorer.

2. As an alternate to URS, we explore similarity between semantic embeddings of question and table row for estimating relevance of tokens. Precisely, to perform this ablation, we encode the question and each row in the table with BERT. Subsequently, we compute the similarity score between the encoding of each table row and question encoding through cosine similarity. Each token in a given table row is then assigned a relevance score equal to the cosine similarity between the row and question encodings. This relevance score is then used (instead of URS) with CABINET. Clearly from 8, replacing URS with BERT-based similarity results in sub-optimal performance when compared against CABINET with URS and without cell highlighter (row 3 vs row 6), hence validating the importance URS.

3. Further, we experiment with another configuration where the weakly supervised cell highlighter is used with off the shelf BERT based encoding similarity for relevance scoring and compare the performance with CABINET (with URS and cell highlighter). Here, a similar trend is observed where it is seen that CABINET with URS and cell highlighter performs much better than CABINET with BERT based relevance scoring and cell highlighter (row 4 vs row 7).

4. Utilising the question directly to generate the highlighted cells, it can be observed that there is a significant decline in performance for the task of TableQA compared to using the parsing statement as input to the cell highlighter to generate the relevant cells which is then used to determine relevance score for table tokens for performing QA (row 5 vs row 7). This indicates that the parsing statement describing the criteria of rows and columns relevant to the question is essential to achieve the performance gains.

Table 8: Performance analysis for different design choices explored over CABINET. OmniTab as a baseline (row 1), ablation on using parsing statement as input to QA LLM instead of higlighting corresponding cells (row 2), leveraging BERT based relevance scoring (row 3 and 4), directly using question as input to cell higlighter LLM (row 5), produce sub-optimal performance when compared with CABINET.

| Method | WikiTQ | FeTaQA | WikiSQL |
|---|---|---|---|
| OmniTab | 63.1 | 35.9 | 85.8 |
| CABINET w parsing statement as input to QA model instead of highlighting corresponding cells | 66.2 | 34.9 | 85.9 |
| CABINET with BERT based relevance scoring (as discussed above) without cell highlighter | 61.8 | 34.9 | 83.7 |
| CABINET with BERT based relevance scoring (as discussed above) with cell highlighter | 64.5 | 36.7 | 85.1 |
| CABINET with question as input to cell highlighter | 63.7 | 34.4 | 85.7 |
| CABINET with URS only and without cell highlighter | 65.6 | 35.8 | 89.3 |
| **CABINET** | **69.1** | **40.5** | **89.5** |

## A.9 DATASET STATISTICS

In this section, we tabulate the number of samples in the training, validation and test set of all the three datasets in table 9.

## A.10 CASE STUDY ON HOW CLUSTERING LOSSES INTERACT TO YIELD IMPROVEMENTS

We now discuss a case study depicting how the loss functions interact to yield improvements:

Table 9: Dataset Statistics

| Dataset | # Train samples | # Validation samples | # Test samples |
|---------|-----------------|----------------------|----------------|
| WikiTQ  | 11321           | 2831                 | 4344           |
| WikiSQL | 56355           | 8421                 | 15878          |
| FeTaQA  | 7326            | 1001                 | 2003           |

Consider the example in figure 2 in the paper pdf - we plot the histogram of relevance score assigned to table tokens during inference for this example for 4 differently trained variants of CABINET - a) URS trained w/o any clustering losses, b) URS trained with clustering loss, c) URS trained with clustering and cluster mean separation loss, and d) URS trained with clustering, cluster mean separation and relevance score sparsification loss. The histogram plots depicting count of table tokens against relevance score assigned during inference for the 4 variants are shown in figure 7, we summarise our observations below:

1. From the sub-plot figure 7(a), it can be seen that when URS is trained without any of the three losses, the relevance score for most of the tokens is in the range 0.7 - 0.9 which is undesirable since many tokens which are irrelevant are also assigned a decently high relevance score which is roughly equivalent to passing the table to the QA LLM as it is.

2. When URS is trained with clustering loss (sub-plot figure 7(b)), the frequency distribution of relevance scores becomes bi-modal with the majority of tokens corresponding to the first mode assigned a relevance score in the range 0.6-0.7 and for the second mode around 0.8-0.9. This shows that clustering loss trains the URS in structuring the latent space representation into two categories, however, still there are many tokens with a relevance score around 0.7 between the two modes which means that many irrelevant tokens are still assigned decently high relevance.

3. When URS is trained with clustering and cluster mean separation loss (figure 7(c)), the first mode corresponding to a relatively lower relevance score is observed around 0.55-0.65 while the second mode corresponding to a higher relevance score is observed around 0.8-0.9. This shows that URS trained with cluster mean separation loss amplifies the effect of clustering loss by pushing more tokens into lower relevance category resulting in more tokens assigned lower relevance around the first mode. Also note that the number of tokens in the range 0.7-0.8 also reduces owing to the increased gap between token representations belonging to two categories.

4. Finally when URS is additionally trained with relevance score sparsification loss (figure 7(d)), we observe that magnitude of relevance score assigned to tokens in low relevance score cluster decreases further which is useful since irrelevant tokens gets suppressed further and the downstream QA LLM can focus better on relevant parts. Further, more irrelevant tokens are assigned a low relevance score. This can be seen in figure 7(d) where more tokens are assigned relevance score in the range 0.45-0.6.

Further, since the URS is trained end-to-end with QA LLM differentiably, tokens in higher score category are likely to be relevant in order to enable the QA LLM to be able to generate the correct answer. Additionally, in figure 8, we visualise the t-SNE plots (corresponding to the same example in figure 2 as discussed above) of the latent representation of table tokens encoded during inference by URS corresponding to the 4 variants trained differently - a) URS trained w/o any clustering losses (figure 8(a)), b) URS trained with clustering loss (figure 8(b)), c) URS trained with clustering and cluster mean separation loss (figure 8(c)), and d) URS trained with clustering, cluster mean separation and relevance score sparsification losses (figure 8(d)). The table tokens having relevance score greater than the average relevance score (assigned to table tokens in this example) are depicted as relevant tokens (in red) while those tokens having score less than the average relevance are depicted as non-relevant (in blue). It can be seen that URS model trained with clustering loss, cluster means separation loss and relevance score sparsification loss is better able to segregate the table tokens into two categories and assign lower relevance to tokens in one category while assigning higher relevance to tokens in the second category. Further, since the URS is trained end-to-end with QA LLM differentiably, tokens in higher score category are likely to be relevant in order to enable the QA LLM to be able to generate the correct answer.

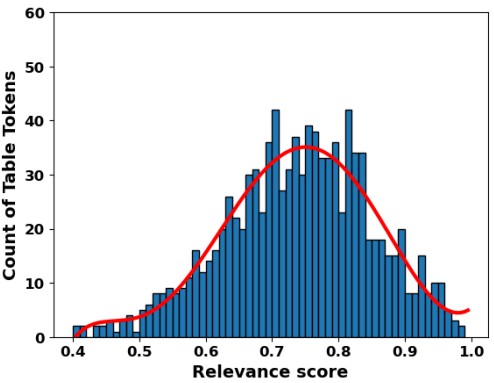

(a) Histogram plot of count of table tokens against relevance score assigned during inference by URS trained without any clustering losses.

(b) Histogram plot of count of table tokens against relevance score assigned during inference by URS trained with clustering loss.

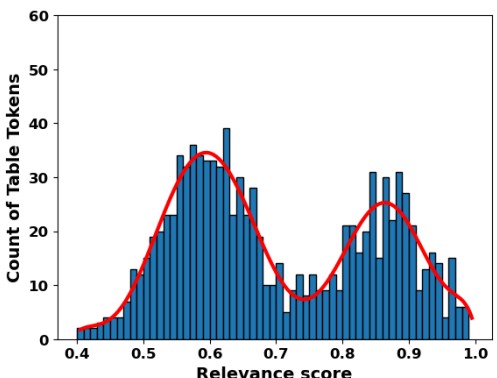

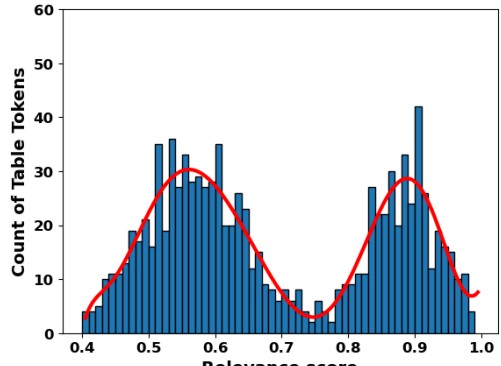

(c) Histogram plot of count of table tokens against relevance score assigned during inference by URS trained with clustering loss and clusters mean separation loss.

(d) Histogram plot of count of table tokens against relevance score assigned during inference by URS trained with clustering loss, clusters mean separation loss and relevance score sparsification loss.

Figure 7: Histogram plots of relevance score assigned during inference to table tokens corresponding to the example in the figure 2 in the main paper. The 4 plots correspond to variants of CABINET trained with different combination of loss functions for URS - a) URS trained w/o any clustering losses; b) URS trained with clustering loss; c) URS trained with clustering and cluster mean separation loss; and d) URS trained with clustering, cluster mean separation and relevance score sparsification loss. It can be seen that the three losses acts in a complementary manner to enable URS to segregate table tokens better into two categories and assign low relevance to tokens in one category and a high relevance to tokens in the second category. Further, since the URS is trained end-to-end with QA LLM differentiably, tokens in higher score category are likely to be relevant in order to enable the QA LLM to be able to generate the correct answer.

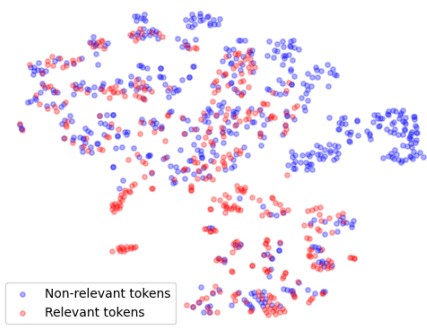

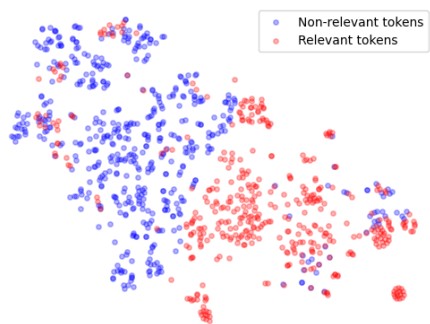

(a) t-SNE plot of latent representation of table tokens encoded during inference by URS trained without any clustering losses.

(b) t-SNE plot of latent representation of table tokens encoded during inference by URS trained with clustering loss only.

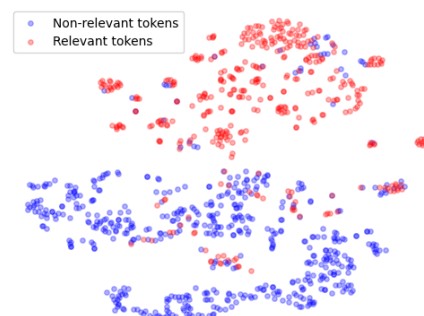

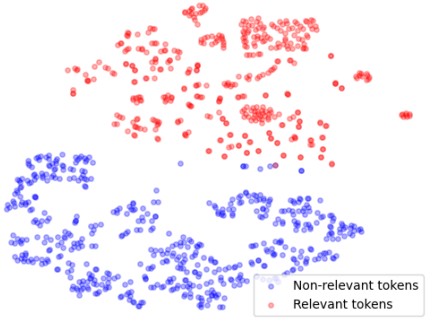

(c) t-SNE plot of latent representation of table tokens encoded during inference by URS trained with clustering loss and cluster means separation loss.

(d) t-SNE plot of latent representation of table tokens encoded during inference by URS trained with clustering loss, cluster means separation and relevance score sparsification loss.

Figure 8: t-SNE plots (corresponding to the same example as in figure 2) of the latent representation of table tokens encoded during inference by URS corresponding to the 4 variants trained differently - a) URS trained w/o any clustering losses (sub-plot figure 8(a)), b) URS trained with clustering loss only (sub-plot figure 8(b)), c) URS trained with clustering and cluster mean separation loss (sub-plot figure 8(c)), and d) URS trained with clustering, cluster mean separation and relevance score sparsification loss (sub-plot figure 8(d)). The table tokens having relevance score greater than the average relevance score (assigned to table tokens in this example) are depicted as relevant tokens (in red) while those tokens having score less than the average relevance are depicted as non-relevant (in blue). It can be seen that URS model trained with clustering loss, cluster means separation loss and relevance score sparsification loss is better able to segregate the table tokens into two categories and assign lower relevance to tokens in one category while assigning higher relevance to tokens in the second category. Further, since the URS is trained end-to-end with QA LLM differentiably, tokens in higher score category are likely to be relevant in order to enable the QA LLM to be able to generate the correct answer.

### A.11 CAN CABINET BE USED FOR IMPROVING OTHER NLP TASKS LIKE READING COMPREHENSION?

We employ CABINET on the reading comprehension task, where given a paragraph and a corresponding question, LLMs need to answer the question based on the paragraph. Drawing an analogy from the Table QA task, certain tokens in the paragraph are more relevant for answering the given question. To achieve this, we employ the URS component of CABINET on top of pre-trained BART-Large. We experiment with a commonly used benchmark SQuAD-v2 and report accuracy on the test set (Table 10), hence validating the task-level generality of CABINET. It can be seen that using CABINET helps in improving the performance of vanilla BART-large LLM for the reading comprehension task.

Table 10: Performance improvement achieved upon employing URS of CABINET over BART for the Reading Comprehension task on SQuAD-v2

| Method | SQuAD-v2 Test Accuracy |
|---|---|
| BART-large | 42.9 |
| **URS of CABINET + BART-large** | **48.0** |

