# OpenReview forum: "CABINET: Content Relevance-based Noise Reduction for Table Question Answering"
_ICLR.cc/2024/Conference — ICLR 2024 spotlight_

### Official Review · Reviewer_mwZa · 2023-10-31

**Soundness:** 3 good
**Presentation:** 3 good
**Contribution:** 3 good
**Rating:** 8
**Confidence:** 4

**Summary:**

The authors propose CABINET (Content RelevAnce-Based NoIse ReductioN for TablE QuesTion-Answering) – a framework to enable LLMs to focus on relevant tabular data by suppressing extraneous information. CABINET comprises an Unsupervised Relevance Scorer, trained differentially with the QA LLM, that weighs the table content based on its relevance to the input question before feeding it to the question-answering LLM (QA LLM). Further, it uses a weakly supervised module that generates a parsing statement describing the criteria of rows and columns relevant to the question and highlights the content of corresponding table cells. Authors release code and dataset to enable for reproducibility.

**Strengths:**

The paper is technically sound and in general well written. Figure 2 is also informative.

**Weaknesses:**

- The authors need to better motivate what the practical utility of Table QA is, as most of the questions are based in the format of natural language text sentences. Moreover, tabular dataset can be indexed and converted to their freeform text representations which general open-domain QA systems are already able to solve.

- In the Related Work Section, the authors should also discuss recent developments with LLM guided graph neural network (GNN) model for QA to capture tabular/graph structure:

[1] EACL 2023: Question-Answer Sentence Graph for Joint Modeling Answer Selection. In Proceedings of the 17th Conference of the European Chapter of the Association for Computational Linguistics, pages 968–979, Dubrovnik, Croatia. Association for Computational Linguistics.
[2] AAAI 2020: TANDA: Transfer and Adapt Pre-Trained Transformer Models for Answer Sentence Selection. AAAI 2020: 7780-7788

- It would be helpful if the authors could better summarize the statistics of their datasets in tabular as opposed to text format. Further, based on the text description, I have a concern that the datasets being used are relatively small-scale, e.g., in the order of thousands of nodes. The experiment results would be more conclusive if evaluated on large-scale datasets. To this end, the authors also should provide analysis on the runtime and memory complexities of their work, since M/B parameter LLMs may not be scalable for training time.

**Questions:**

Please see weaknesses section above

---

> ### Author Response · Authors · 2023-11-16
> **Author Response to Reviewer mwZa (1/2)**
>
> Thank you for your encouraging review and constructive feedback! We provide our response below:
> ***
>
> > "The authors need to better motivate what the practical utility of Table QA is, as most of the questions are based in the format of natural language text sentences."
>
> Table QA is a problem of immense practical utility. Much information is generated, transmitted, and stored in tabular format (e.g., financial documents, enterprise reports, documents, summarized statistics and trends in newspapers and magazines, etc.). Table QA models aim to unlock this vast information by parsing and reasoning over tabular data, and enabling end-users to ask questions about tabular data in natural language, and gain insights about the data in the table conveniently by asking questions without having to go through large amounts of data in the table manually.
>
> ***
>
> > "Moreover, tabular dataset can be indexed and converted to their freeform text representations which general open-domain QA systems are already able to solve."
>
> Indexing and searching over free-from text representations of tabular data is an intuitive first solution. Many Table QA systems, including the proposed CABINET, do utilize these text representations (or verbalizations, as we call them in the paper) in some form. However, these textual representations are insufficient due to the complexity of tabular data (diverse formats and structures, row and column positions, numerical data, etc.). Further, simply retrieving verbalized information can not answer questions that require complex reasoning and aggregation operations -- an area where the proposed CABINET especially shines (Appendix A4).
>
> Further, do note that the SoTA QA systems (open-domain, or tabular) are based on GPT scale LLMs and are typically tens of billions of parameters in size. As discussed in the paper (section 4.1), CABINET significantly outperforms methods built on top of such LLMs despite having orders of magnitude fewer parameters (CABINET contains 560M parameters while models built on ChatGPT/GPT-3 contain 175B parameters).
>
> ***
> > "In the Related Work Section, the authors should also discuss recent developments with LLM guided graph neural network (GNN) model for QA to capture tabular/graph structure"
>
> Thank you for your suggestion. These works are certainly relevant and we have included the discussion regarding them in the revised draft (in the related work section in the main paper and Appendix A.3).
>
> ***
>
> > "It would be helpful if the authors could better summarize the statistics of their datasets in tabular as opposed to text format."
>
> Thanks, good suggestion! We had written the statistics of datasets in text format due to the paucity of space in the main paper. We have now modified the pdf of the paper to summarise the dataset stats in a tabular format in Appendix A.9 (Table 9).
>
> ***
>
> > "Further, based on the text description, I have a concern that the datasets being used are relatively small-scale, e.g., in the order of thousands of nodes. The experiment results would be more conclusive if evaluated on large-scale datasets."
>
> Regarding the scale of the datasets, we selected the most commonly used publicly available benchmarks for evaluating CABINET that allowed us to compare CABINET uniformly and fairly with other SoTA methods which have used the same datasets for Table QA. The correct way to judge the scale of the datasets, we would argue, is in terms of number of questions and the complexity of the questions. All three datasets provide tens of thousands of questions, with WikiSQL comprising 80K+ questions over 24K tables! Further, FetaQA, which comprises 10K+ questions, requires fetching multiple entities from the table, aggregating and reasoning over these entities, and structuring the inferred information to produce a long, coherent, natural language answer (Section 4). Do also note that each table in many cases contains hundreds of cells (Section 4.3 - impact of table size on performance). Thus, the datasets chosen are among the hardest, publicly available datasets for the task.
>
> ***

---

> ### Author Response · Authors · 2023-11-16
> **Author Response to Reviewer mwZa (2/2)**
>
> ***
>
> > "To this end, the authors also should provide analysis on the runtime and memory complexities of their work, since M/B parameter LLMs may not be scalable for training time"
>
> We are completely aligned with efficiency concerns. That’s why CABINET contains only 560M parameters (which comes under the very lightweight LLMs category), and despite that, it is able to outperform much larger ChatGPT/GPT-3 based DATER containing 175B parameters. Hence, at a time when the general trend is to increase the number of parameters, CABINET offers a carefully designed solution keeping practical utility in mind and achieves significantly better performance efficiently with orders of magnitude fewer parameters.
>
> * The average runtime time required during inference for CABINET (averaged over 100 queries) is: 0.212s
> * The average runtime required during inference for one of the best-performing baselines - OmniTab is (averaged over 100 queries): 0.153s
> * The GPU memory required for CABINET during inference (comprising 560M parameters) is: 2.86 GB
> * The GPU memory required for OmniTab during inference (comprising 440M parameters) is: 2.27 GB
> * The GPU memory required for CABINET during training per gpu with batch size 8 (comprising 560M parameters) is: 54 GB
> * The GPU memory required for OmniTab during training per gpu with batch size 8 (comprising 440M parameters) is: 36 GB
>
> It can be noted that CABINET gives large performance gains at the cost of small incremental increase in inference latency.

---

> > ### Comment · Reviewer_mwZa · 2023-11-23
> >
> > Thanks to the authors for providing a response to the questions raised. In light of the detailed responses, this is a good paper and I have raised my score of the paper.

---

> > > ### Author Response · Authors · 2023-11-23
> > >
> > > Thank you for acknowledging the responses and updating the score. We appreciate your encouraging and helpful feedback.

---

### Official Review · Reviewer_e8r4 · 2023-10-31

**Soundness:** 3 good
**Presentation:** 3 good
**Contribution:** 3 good
**Rating:** 8
**Confidence:** 4

**Summary:**

This paper tackles the task of table Question-Answering (QA). The authors state that the approaches considering all the tables have too much noise when generating the answer, while approaches selecting a part of the table before answering might remove the relevant parts. The authors thus propose to estimate the relevance of each token based on a clustering approach with latent variable (Srivastava and Sutton, 2017), coupled with two custom losses (sparsity of the relevance cluster and distance between the centroids). They combine this token relevance score with a predictor of a table "cell relevance" based on a sequence-to-sequence model that outputs the relevant cells from a full table.

Results reported on the main table QA datasets (WikiTableQuestion, FeTaQa, and WikiSQL) show a noticeable improvement (e.g. gaining 3 points in accuracy compared to a LLM/Codex prompting approach that select subtables, DATER, on WikiTQ). Experiments consisting in perturbing the tables (adding rows, column/row permutation, cell replacement) show that the method is more robust than OmniTAB and ReasTAP. Finally,

**Strengths:**

- The approach has good results on the main table QA dataset
- The ablation study shows that each subcomponent of the model is important
- The model outperforms LLM-based approaches with a much lower number of parameters (175B vs 3.5B) and other state-of-the-art approaches (e.g. OmniTAB)

**Weaknesses:**

- The overall model is quite complex, relying on various subcomponents of different natures: this paper provides a new baseline, but it is hard to build from that
- The cell relevance loss is not learned, making the model not end-to-end
- Using a latent variable for relevance, which is optimized as within a probabilistic model for relevance, but then used as a scalar, makes the model a bit inconsistent - a fully probabilistic model would have been much sounder
- there are missing experimental details

**Questions:**

- section 3.1, the whole paragraph discussing variational inference sounds strange: variational inference is a method to estimate laternt variable probabilities - stating that "we model ... as a latent variable ... **through VI**" is not correct

- the experimental details (how the model was trained, how hyperparameters were set) are not given; could the authors provide the information (at least in the appendix)?

- How is the model used during inference (i.e. how is the VI-related loss used to predict the relevance of each token)?

- is the clustering model that important? Would predicting directly a probability of relevance for each token work?

- p.9, please correct "its performance" to "TAPEX performance" for clarity

---

> ### Author Response · Authors · 2023-11-16
> **Author Response to Reviewer e8r4 (1/2)**
>
> Thank you for your encouraging review and constructive feedback! We provide our response below:
>
> ***
>
> > "The overall model is quite complex, relying on various subcomponents of different natures ... The cell relevance loss is not learned, making the model not end-to-end"
>
> LLMs are sensitive to noise in the input. Hence, one major aspect of the proposed CABINET framework is the ability to identify relevant parts of the table to the input question. This reduces the "noise" in the input to the QA LLM and it can focus on the relevant information for generating the correct answer.
>
> Note that in practical deployment settings, such fine-grained annotations denoting parts of the table relevant to a question are not available. Keeping this constraint in mind, we introduce an unsupervised relevance scorer component trained in an end-to-end manner with the question-answering LLM. Next, to further capture the fine-grained relevance signals, we propose a weakly supervised cell predictor module to highlight relevant cells of the table conditioned on a statement describing how the table should be parsed to answer a given question. Again, owing to the unavailability of fine-grained table data annotations (obtaining which is a very costly process), we make this deliberate design choice of using unsupervised and weakly supervised components to tackle the problem of noisy table tokens and providing highly relevant signals to the QA LLM.
>
> ***
>
> > "section 3.1, the whole paragraph discussing variational inference sounds strange: variational inference is a method to estimate latent variable probabilities - stating that "we model ... as a latent variable ... through VI" is not correct"
>
> Thank you for the suggestion! We agree we could have phrased this better. Recall that we aim to predict the relevance of each table token. However, since ground-truth annotations for relevant table tokens are unavailable, table token relevance is unobserved, and we consider it a latent variable. As you mentioned, variational inference is a method to estimate latent variable probability, and we use it to estimate token relevance probability. Accordingly, we have now rephrased the paragraph in the updated paper pdf (section 3.1, page 4) as follows:
>
> “We aim to predict relevance score for each table token, however, since annotations for relevant table parts are unavailable, token relevance is not explicitly observable and we consider it as a latent variable. Further, we hypothesize that the representation space of table tokens can be structured better for modeling relevance by clustering their encodings into two categories - relevant and non-relevant. Variational Inference (VI) has been commonly used to estimate latent variable probability and group data points on the basis of latent topics. Hence, we estimate relevance of table token as (step 3 in Fig.2): ...”
>
> ***
>
> > "is the clustering model that important? Would predicting directly a probability of relevance for each token work?"
>
> Yes, the clustering model is important. Clustering the table tokens using their latent representation enables CABINET to separate the relevant tokens for the question from the non-relevant ones. As the ablation reported in Table 4 (in the main paper, section 4.4) reveals, settings, where clustering is performed, achieve higher performance compared to settings where clustering is not performed. For instance, in row 6 - clustering-related losses lead to ~4.5% better acc. on WikiTQ, 0.7% better on FeTaQA, and ~3% better performance on WikiSQL, compared to row 1 (no-clustering).
>
> Please refer to Appendix A.10 and figures 7-8 (pages 21-23) in the updated pdf for a case study depicting the importance of clustering and how the different losses interact with each other to yield improvement.
>
> ***

---

> > ### Comment · Reviewer_e8r4 · 2023-11-21
> >
> > Thanks for your answers and the clarification on variational inference - the paragraph is much clearer like this. Given the answers given to other concerns raised by other reviewers, I agree to rate this paper higher since it improves over state-of-the-art and proposes a latent clustering approach that helps the task at hand. The only downside is the cell relevance tackled using an LLM that makes the model more "hacky" – but this is necessary to reach state-of-the-art and could be addressed in future works.

---

> > > ### Author Response · Authors · 2023-11-22
> > >
> > > Thank you for reviewing our responses, updating the score, and providing valuable feedback.

---

> ### Author Response · Authors · 2023-11-16
> **Author Response to Reviewer e8r4 (2/2)**
>
> ***
>
> > "How is the model used during inference (i.e. how is the VI-related loss used to predict the relevance of each token)?"
>
> The VI-related losses (clustering loss, cluster mean separation loss, and relevance score sparsification loss) are only applied during training to tune the weights of the unsupervised relevance scorer to assign appropriate relevance scores to table tokens for a given question. Hence, these losses are not (required to be) applied during inference to estimate relevance scores. Applying these losses together at the training time (as a training signal) to optimize the learnable parameters of the unsupervised relevance scorer through gradient backpropagation allows it to learn to assign a high score to relevant tokens and a low score to non-relevant tokens to enable the downstream QA model to generate the right answer.
>
> For better clarity, we explain the inference flow to compute the relevance score as follows: As discussed in section 3.1, the linearised table and the question are given as input to the unsupervised relevance scorer which is a transformer encoder (eqs.1-2). The latent representation (eq.3) encoded by this transformer encoder corresponding to each table token is then used to estimate the relevance logit through variational inference (eqs.4-5) such that sigmoid is applied over this logit to obtain the relevance score.
>
> ***
>
> > "the experimental details (how the model was trained, how hyperparameters were set) are not given; could the authors provide the information (at least in the appendix)?"
>
> Yes, certainly! We have elaborated more implementation details for CABINET and added them to the appendix A.6. Details on how table perturbation is performed was present in appendix A.5. The full implementation details (in section 4 plus additional details) added to appendix A.6 are as follows:
>
> As stated in section 4 of the main paper (under implementation details), we employ the OmniTab backbone built over BART-Large architecture for the encoder ($TE_{QA}$) and decoder ($TD_{QA}$) of the QA LLM. Further, URS encoder ($TE_{URS}$) is initialized with the architecture and weights of QA LLM encoder ($TE_{QA}$), though, they do not share weights during training. However, the embedding layers $Embedding_{URS}$ and $Embedding_{QA}$ share the weights. The hidden dimension $d$ of $TE_{URS}$ is $1024$, similar to that of both $TE_{QA}$ and $TD_{QA}$. We train CABINET and other baselines for 30 epochs on an effective batch size (BS) of 128 using 8 80 GB A100 GPUs (BS of 8/GPU with gradient accumulation 2) using learning rate of $1e^{-5}$ with Cosine Annealing through AdamW optimizer. We carry out exhaustive hyper-parameter tuning based on the validation set to come up with the optimal values of learning rate ($1e^{-5}$), scheduler (Cosine Annealing), batch size (8), gradient accumulation steps (2) and the optimizer (AdamW). We leverage text pre-trained Flan T5-xl as the backbone for Parsing Statement Generator (PSG) and the Cell Highlighter LLM. The Parsing Statement Generator is trained for 50 epochs on the $~300$ sampled question-table-parsing statement triplets with hyper-parameters setting same as that of CABINET, however the only difference being the effective batch size of 16 (BS of 2/GPU with 8 gpus without gradient accumulation). On a similar note, the Cell Highlighter LLM is trained for 3 epochs on the ToTTo dataset to generate highlighted cells given the table and a natural language statement describing a subpart of the table as the input. During the inference phase of generating answer, parsing statements and highlighted cells, we use beam search decoding with a beam size of 3.
>
> ***
>
> > "p.9, please correct "its performance" to "TAPEX performance" for clarity"
>
> Thanks for suggesting the correction! We have updated this in the paper pdf.
>
> ***

---

> ### Author Response · Authors · 2023-11-19
> **(Contd.) Author Response to Reviewer e8r4**
>
> ***
> > "this paper provides a new baseline, but it is hard to build from that"
>
> We believe that our method is generic and can be applied in other NLP tasks (like reading comprehension) to improve the performance of an existing model to enable it to focus on relevant parts of the input. To illustrate our method's applicability, broader impact and how other methods can be built on one of its core components in a plug-and-play manner, we experimented with applying the URS (Unsupervised Relevance Scorer) component of CABINET on the well-known SQuAD-2.0 dataset commonly used for machine reading comprehension task.
>
> In this task, given a passage and a related question, the model has to generate an answer to the question by extracting relevant information from the passage. We train a BART-Large-based generative LLM as a baseline that takes the passage and question as input and is trained to generate the answer using the samples in the training dataset. Separately, we train another model where we employ URS (trained with clustering loss, cluster means separation loss, and score sparsification loss) with the BART-Large model as described in the paper to assign a relevance score to each token in the passage given the question and passage as input. At test time, the generated answer is compared against the expected answer to estimate exact match accuracy.
>
> We summarise the accuracy obtained in the following table where it can be seen that using CABINET's URS with the BART model helps boost accuracy by **~5.1%** by suppressing extraneous information in the passage that is not relevant to the question and enabling the BART model to focus on relevant parts of the passage to generate the correct answer.
>
> | Method  | Test Accuracy (%)   |
> | ---------- | ---------- |
> | BART-Large | 42.85 |
> | BART-Large with URS (of CABINET) | 47.96 |
>
> Additionally, to enable the research community to use our method and build on it for different tasks, we have already released the code for CABINET with the link mentioned in the abstract.
>
> We mention further implementation details of the reading comprehension experiments here for completeness - The number of samples in the training and test set of SQuAD-2.0 are 130k and 11.9k respectively. We trained both the methods i.e. baseline BART-Large and BART-Large with URS for 30 epochs with a learning rate of $1e^{-5}$, effective of batch size of  128 (BS 8 / GPU with 8 GPUs and gradient accumulation 2), optimized using AdamW with Cosine Annealing learning rate scheduler.
>
>
> ***

---

### Official Review · Reviewer_nyxN · 2023-11-01

**Soundness:** 4 excellent
**Presentation:** 4 excellent
**Contribution:** 4 excellent
**Rating:** 8
**Confidence:** 4

**Summary:**

This paper introduces a novel architecture for table QA. It is with two components, which are the main contributions of this paper. Unsupervised Relevance Scorer provides a "soft" assignment score for each row of the table given the question. Relevant Cell Predictor through Table Parsing converts the question into a statement with highlighted column and row information that can be used to solve the QA tasks. This method achieves SOTA performances on WikiTable, FeTaQA, and WikiSQL. Overall, this is a great paper.

**Strengths:**

* This paper proposed a novel framework for table QA tasks
* The proposed method achieves SOTA performances on three table QA tasks
* The paper shows the robustness of the provided model

**Weaknesses:**

* Unlike a "hard" table token retriever, the computation of the model proposed by this paper may be extensive when the number of table tokens is many.
* More ablation studies are needed to support the needs of the components introduced in this paper.

**Questions:**

* What is the average model inference latency compared with the baseline model (OmniTAB) for the three tasks
* More ablation studies are needed to support the needs of the components introduced in this paper.
  - The need for an Unsupervised Relevance Scorer: If we replace URS with BM25 or some out-of-the-box similarity metrics, will the model have a huge performance decline?
  - The need for a Parsing Statement Generator: If the original question is directly used for cell highlighter other than the parsed text, will the model have a huge performance decline?
  - The need for a Cell Highlighter: If the $\eta^{cell}$ is the indicator for a cell value in the parsed text, will the model have a huge performance decline?

---

> ### Author Response · Authors · 2023-11-16
> **Author Response to Reviewer nyxN**
>
> Thank you for the encouraging words and helpful suggestions for additional ablations. Please find our responses and results of the requested experiments below. We have also added the results for the suggested experiments in the revised PDF of the paper (Appendix A.8, Table 8).
>
> ***
>
> > "What is the average model inference latency compared with the baseline model (OmniTAB)"
>
>
> * The average runtime time required during inference for CABINET (averaged over 100 queries) is: 0.212s
> * The average runtime required during inference for one of the best-performing baselines - OmniTab is (averaged over 100 queries): 0.153s
>
> It is to be noted that the inference time for the model remains the same irrespective of the three datasets. It can be seen that CABINET gives large performance gains at the cost of small incremental increase in inference latency.
>
> ***
>
> > "The need for an Unsupervised Relevance Scorer: If we replace URS with BM25 or some out-of-the-box similarity metrics, will the model have a huge performance decline?"
>
> To answer this question, we did the following experiment: We encode the question and each row in the table with BERT. Subsequently, we compute the similarity score between the encoding of each table row and question encoding through cosine similarity. Each token in a given table row is then assigned a relevance score equal to the cosine similarity between the row and question encodings. This BERT encoding similarity based relevance score is then used (instead of URS) with CABINET. We summarise the results obtained in the following table and compare them with CABINET:
>
> | Method   | WikiTQ (Acc.)   | FeTaQA (SacreBleu)   | WikiSQL (Acc.)   |
> | ---------- | ---------- | ---------- | ---------- |
> | CABINET with BERT similarity based relevance scoring (as discussed above) without cell highlighter | 61.8 | 34.9 | 83.7 |
> | CABINET with URS only and without cell highlighter | 65.6 | 35.8 | 89.3 |
>
> As can be noted, replacing URS in CABINET with an off-the-shelf BERT encoding-based similarity for relevance scoring results in significant drop in performance. Next, we also experiment with another configuration where the weakly supervised cell highlighter is used with BERT encoding-based similarity metrics for relevance scoring and compare the performance with CABINET (with URS and cell highlighter) in the following table. Here, a similar trend is observed where it is seen that CABINET with URS and cell highlighter performs much better than CABINET with BERT encoding similarity based relevance scoring and cell highlighter.
>
> | Method   | WikiTQ (Acc.)   | FeTaQA (SacreBleu)   | WikiSQL (Acc.)   |
> | ---------- | ---------- | ---------- | ---------- |
> | CABINET with BERT encoding similarity based relevance scoring (as discussed above) with cell highlighter | 64.5 | 36.7 | 85.1 |
> | CABINET with URS and cell highlighter | 69.1 | 40.5 | 89.5 |
>
> ***
> ***
>
> > "The need for a Parsing Statement Generator: If the original question is directly used for cell highlighter other than the parsed text, will the model have a huge performance decline?"
>
> The following table summarises the TableQA performance achieved when the question is used as input to the cell highlighter instead of the parsing statement.
>
> | Method   | WikiTQ (Acc.)   | FeTaQA (SacreBleu)   | WikiSQL (Acc.)   |
> | ---------- | ---------- | ---------- | ---------- |
> | CABINET with question as input to cell highlighter | 63.7 | 34.4 | 85.7 |
> | CABINET with parsing statement as input to cell highlighter | 69.1 | 40.5 | 89.5 |
>
> It can be observed that there is a significant decline in accuracy when question is given as input to the cell highlighter compared to giving the parsing statement as input to the cell highlighter. This indicates that the parsing statement describing the criteria of rows and columns relevant to the question is essential to achieve the performance gains.
>
> ***
>
> > "The need for a Cell Highlighter: If the $\eta^{cell}_{p}$ is the indicator for a cell value in the parsed text, will the model have a huge performance decline?"
>
> We perform an ablation where tokens corresponding to table cell values which are present in the parsing statement text are highlighted instead of using the cell highlighter model predictions and compare with CABINET in the following table where it can be seen that the performance drops significantly compared to CABINET. Thus, the cell highlighter model is needed to achieve performance gains to predict relevant cells corresponding to the parsing statement.
>
> | Method   | WikiTQ (Acc.)   | FeTaQA (SacreBleu)   | WikiSQL (Acc.)   |
> | ---------- | ---------- | ---------- | ---------- |
> | CABINET with table cell values present exactly in parsing statement text used as highlighted cells| 65.4 | 36.9 | 86.2 |
> | CABINET with cell highlighter model used as described in the paper | 69.1 | 40.5 | 89.5 |
>
> ***

---

> > ### Comment · Reviewer_nyxN · 2023-11-19
> >
> > Thanks for the response. I read the reply and it does address my previous concerns about the latency and missing of some ablations. According to the reply, the latency increase is reasonable. Also so is GPU memory increase in the reply to reviewer mwZa. I really appreciate that the authors showed the ablation study numbers quickly in the reply. The baselines are reasonable and the ablation study shows the three components are necessary. Since my previous rating is high, I will keep my previous rating.

---

> > > ### Author Response · Authors · 2023-11-20
> > >
> > > Thank you for reading our response and providing encouraging feedback.

---

### Official Review · Reviewer_NUWR · 2023-11-02

**Soundness:** 3 good
**Presentation:** 2 fair
**Contribution:** 3 good
**Rating:** 8
**Confidence:** 5

**Summary:**

To minimize the negative effects from the noisy and distracting information from irrelevant parts of the table, in this work, the authors propose CABINET framework that weighs different table parts based on their relevance to the question without explicitly removing any content. The main part of this framework is on the relevance score assignment, either with unsupervised learning or leveraging external weakly-supervised cell highlighter model, which provides the relevance measurement between table content and question.
Specifically, variational inference is used to obtain the relevance score with unsupervised fashion. Furthermore, CABINET leverages a parsing statement generator that describes which rows and columns are relevant to the question to provide more matching information. Experimental results show that: (1) The relevance score with three designed loss functions can improve the model performance. (2) The relevance scores obtained from unsupervised setting and weakly-supervised setting should be balanced to obtain good performance.

**Strengths:**

The model achieves the state-of-the-art performance on the WikiTQ, FetaQA and WikiSQL using orders of magnitude fewer parameters when compared of LLMs such as Codex.
Furthermore, the model is more robust on the perturbation than other counterparts. Also for large tables, the CABINET also shows better performance than OmniTab.
The authors smartly use the ToTTo dataset to train a cell highlighter model for obtaining the relevance score from the parsing statement.
The ~300 manually annotated example for parsing statement generation will be very useful for the community.

**Weaknesses:**

Even though there is a good ablation experiments on the three loss functions, my overall feeling is that there is no good rational how these loss function interact with each other. For example, we can see that combining all three loss functions performs best. However, the benefits of a single loss function is hard to be observed. I think this strategy can be further applied to reading comprehension task. If it works, this can make the paper more comprehensive, having broader impact. No other weaknesses in my mind but there are some technical questions in Question section.

-----
With responses and updated number for reading comprehension task, I am happy to change the rating accordingly.

**Questions:**

What’s the performance of cell highlighter on ToTTo validation set? Do we have evaluations on some sampled examples from FetaQA, WikiTQ, and WikiSQL showing the quality of parsing statement generator? If the quality of parsing statement is high but putting more weight on that makes the model perform worse, then it would be an interesting point to discuss.
From Table 4, there is a big improvement of using all three loss functions against the models without using all three loss functions. Do we have some case studies showing where the improvements come from?

---

> ### Author Response · Authors · 2023-11-16
> **Author Response to Reviewer NUWR (1/2)**
>
> Thank you for your careful read of our paper and insightful questions. Please find our answers and clarifications below.
>
> ***
>
> > "What’s the performance of cell highlighter on ToTTo validation set? ... If the quality of parsing statement is high but putting more weight on that makes the model perform worse, then it would be an interesting point to discuss."
>
> The accuracy of cell highlighter on ToTTo validation set is 61.81% for the task of predicting the correct set of cells corresponding to the natural language statement (about the table) given as input. Further, the average number of ground truth cells highlighted per sample as the target output label in the ToTTo dataset is 3.78. These empirical observations do lend credence to the argument that the cell highlighter learns to predict fewer and more precise cells corresponding to the text sentence given as input to the cell highlighter. Moreover, while the accuracy of 61.81% for cell highlighter is decent but not very high, and due to the fact that the cell highlighter predicts very few relevant cells owing to the nature of the ToTTo dataset on which it is trained, we posit that giving very high weight to cell highlighter is resulting in masking out the majority of the parts of the table (some of which could be relevant to the question) causing a loss of information and low performance.
>
> However, simultaneously, we would like to emphasize that the cell highlighter helps boost the relevance score for the small number of highly relevant cells of the table and thus, acts in a complementary manner when combined using appropriate weightage through linear combination with the score from the unsupervised relevance scorer to enable the downstream QA model to focus on relevant information in the table leading to overall performance gains.
>
> ***
>
> > "Even though there is a good ablation experiments on the three loss functions, my overall feeling is that there is no good rational how these loss function interact with each other."
>
> We explain the rationale behind using the loss functions. The unsupervised relevance scorer (URS) is trained with the QA LLM differentiably so that it learns to assign appropriate relevance scores to table tokens to enable the QA LLM to focus on relevant parts of the table in order to be able to generate the correct answer. To further aid the training of URS, we:
>
> * Tune the weights of URS by applying clustering loss (at training time) to the latent representation of table tokens computed by URS so that it learns to implicitly encode the table tokens in a way such that they approximately get segregated into relevant and non-relevant categories conditioned on the given question and an appropriate relevance score is assigned to table tokens in each category accordingly.
> * To enforce this further, we apply mean separation loss between the two cluster center vectors so that the latent representations and distribution of relevance scores for the corresponding table tokens in two clusters are well separated and the mean vector of the cluster representing relevant tokens is distanced away from the mean vector of the cluster representing non-relevant tokens.
> * Finally, we apply sparsification loss to ensure that relevance scores for table tokens in one cluster are low (corresponding to non-relevant category) and relatively higher for the tokens in the other cluster (corresponding to relevant category). Further, this is done to ensure that not all table tokens are assigned a high relevance score which would be equivalent to passing the entire table as is to the QA LLM.
>
> ***

---

> ### Author Response · Authors · 2023-11-16
> **Author Response to Reviewer NUWR (2/2)**
>
> ***
>
> > "From Table 4, there is a big improvement of using all three loss functions against the models without using all three loss functions. Do we have some case studies showing where the improvements come from?"
>
> We now discuss a case study depicting the above rationale on how the loss functions interact to yield improvements:
>
> Consider the example in figure 2 in the paper pdf. For this example, we plot the histogram of relevance score assigned to table tokens during inference by 4 variants of CABINET trained differently - 1) URS trained w/o any clustering losses, 2) URS trained with clustering loss, 3) URS trained with clustering and cluster mean separation loss, and 4) URS trained with clustering, cluster mean separation and relevance score sparsification loss. Since it is not possible to add the histogram plots depicting relevance score distribution in the response here, we have added them as figure 7 [a-d] in appendix A.10 (page 21 in the updated pdf) respectively for the 4 variants. We summarise our observations from these figures below:
>
> * From the sub-plot figure 7(a), it can be seen that when URS is trained without any of the three losses, the relevance score for most of the tokens is in the range 0.7 - 0.9 which is undesirable since many tokens which are irrelevant are also assigned a decently high relevance score which is roughly equivalent to passing the table to the QA LLM as it is.
> * When URS is trained with clustering loss (sub-plot figure 7(b)), the frequency distribution of relevance scores becomes bi-modal with the majority of tokens corresponding to the first mode (cluster) assigned a relevance score in the range 0.6-0.7 and for the second mode (cluster) around 0.8-0.9. This shows that clustering loss trains the URS in structuring the latent space representation of table tokens into two categories, however, still there are many tokens with a relevance score around 0.7 between the two modes which means that many irrelevant tokens are still as signed decently high relevance.
> * When URS is trained with clustering and cluster mean separation loss (figure 7(c)), the first mode corresponding to the relatively lower relevance score cluster is observed around 0.55-0.65 while the second mode corresponding to a higher relevance score cluster is observed around 0.8-0.9. This shows that URS trained with cluster mean separation loss amplifies the effect of clustering loss by pushing more tokens into lower relevance category resulting in more tokens assigned lower relevance around the first mode. Also note that the number of tokens in the range 0.7-0.8 also reduces owing to the increased gap between token representations belonging to two categories.
> * Finally when URS is additionally trained with relevance score sparsification loss (figure 7(d)), we observe that magnitude of relevance score assigned to tokens in low relevance score cluster decreases further which is useful since irrelevant tokens get suppressed further and the downstream QA LLM can focus better on relevant parts. Further, more irrelevant tokens are assigned a low relevance score. This can be seen in figure 7(d) where more tokens are assigned relevance score in the range 0.45-0.6.
>
> Further, since the URS is trained end-to-end with QA LLM differentiably, tokens in higher score category are likely to be relevant in order to enable the QA LLM to be able to generate the correct answer.
>
> Additionally, in figure 8 (Appendix A.10 page 23 in updated pdf), we visualise the t-SNE plots (corresponding to the same example in figure 2 as discussed above) of the latent representation of table tokens encoded during inference by URS corresponding to the 4 variants trained differently - a) URS trained w/o any clustering losses (figure 8(a)), b) URS trained with clustering loss (figure 8(b)), c) URS trained with clustering and cluster mean separation loss (figure 8(c)), and d) URS trained with clustering, cluster mean separation and relevance score sparsification losses (figure 8(d)). The table tokens having relevance score greater than the average relevance score (assigned to table tokens in this example) are depicted as relevant tokens (in red) while those tokens having score less than the average relevance are depicted as non-relevant (in blue). It can be seen that URS model trained with clustering loss, cluster means separation loss and relevance score sparsification loss is better able to segregate the table tokens into two categories and assign lower relevance to tokens in one category while assigning higher relevance to tokens in the second category. Further, since the URS is trained end-to-end with QA LLM differentiably, tokens in higher score category are likely to be relevant in order to enable the QA LLM to be able to generate the correct answer.
>
> ***

---

> ### Author Response · Authors · 2023-11-19
> **(Contd.) Author Response to Reviewer NUWR on Applying CABINET's URS for Reading Comprehension Task**
>
> ***
>
> > "I think this strategy can be further applied to reading comprehension task. If it works, this can make the paper more comprehensive, having a broader impact."
>
> As suggested, we experimented with applying the URS (Unsupervised Relevance Scorer) component of CABINET on the well-known SQuAD-2.0 dataset commonly used for machine reading comprehension task. In this task, given a passage and a related question, the model has to generate an answer to the question by extracting relevant information from the passage. We train a BART-Large-based generative LLM as a baseline that takes the passage and question as input and is trained to generate the answer using the samples in the training dataset. Separately, we train another model where we employ URS (trained with clustering loss, cluster means separation loss, and score sparsification loss) with the BART-Large model as described in the paper to assign a relevance score to each token in the passage given the question and passage as input. At test time, the generated answer is compared against the expected answer to estimate exact match accuracy.
>
> We summarise the accuracy obtained in the following table where it can be seen that using CABINET's URS with the BART model helps boost accuracy by **~5.1%** by suppressing extraneous information in the passage that is not relevant to the question and enabling the BART model to focus on relevant parts of the passage to generate the correct answer.
>
> | Method  | Test Accuracy (%)   |
> | ---------- | ---------- |
> | BART-Large | 42.85 |
> | BART-Large with URS (of CABINET) | 47.96 |
>
> We mention further implementation details for the reading comprehension experiments here for completeness - The number of samples in the training and test set of SQuAD-2.0 are 130k and 11.9k respectively. We trained both the methods i.e. baseline BART-Large and BART-Large with URS for 30 epochs with a learning rate of $1e^{-5}$, effective of batch size of  128 (BS 8 / GPU with 8 GPUs and gradient accumulation 2), optimized using AdamW with Cosine Annealing learning rate scheduler.
>
> ***

---

### Author Response · Authors · 2023-11-20
**General Author Response**

We would like to thank the reviewers again for their constructive reviews. We summarise the additional experiments performed during the discussion phase based on reviewers’s suggestions and corresponding observations in the following points:

* To illustrate our method's applicability, broader impact, and how it can be used in a plug-and-play manner, we experimented with applying the URS (Unsupervised Relevance Scorer) component of CABINET on the well-known SQuAD-2.0 dataset commonly used for machine reading comprehension task. In this task, given a passage and a question related to the passage, the model has to use relevant information in the passage to generate the answer to the question. We observe a performance gain of ~5% achieved through using URS with BART-Large compared to the vanilla BART-Large baseline. This illustrates that URS enables the LLM to focus on relevant parts of the passage to generate the correct answer.

* We carry out additional ablation studies to justify our design choices for CABINET. We experimented with the following ablation variants - 1) using text tokens in the parsing statement that are exactly present in the table cells to determine the cell relevance score $\eta^{cell}_{p}$ instead of using the cell highlighter model; 2) replacing URS with off-the-shelf BERT encoding-based similarity metric between the question and table content for relevance scoring; and 3) using question directly (instead of parsing statement) as input to the cell highlighter model to generate highlighted cells. It is observed that all these variants provide substantially lesser performance compared to CABINET.

* We provide a case study (in Appendix A.10, pages 21-23) depicting how clustering loss, cluster means separation loss and relevance score sparsification loss interact to yield improvements. The histogram plots (Figure 7, Appendix A.10, page 21) of relevance score assigned to table tokens for an example sample showcase that the three losses act in a complementary manner to enable URS to segregate table tokens better into two categories and assign a low relevance score to tokens in one category and a high relevance score to tokens in the second category (please refer to Appendix A.10 on page 21-22 for further elaboration). Further, since the URS is trained end-to-end with QA LLM differentiably, tokens in the higher score category are likely to be relevant for the QA LLM to be able to generate the correct answer.
Additionally, we plot the t-SNE plots (please see figure 8, Appendix A.10, page 23) of latent representations of table tokens encoded by URS where a similar observation holds such that the URS model trained with these losses is better able to segregate the table tokens into two categories and assign lower relevance to tokens in one category while assigning higher relevance to tokens in the other category.

---

### Meta-Review · Area_Chair_zbYe · 2023-12-10

**Metareview:**

The paper introduces a framework for table question-answering (QA) that aims to reduce noise by focusing on relevant tabular data. The proposed CABINET framework consists of an Unsupervised Relevance Scorer (URS) and a weakly supervised module for generating parsing statements. The paper shows state-of-the-art performance on WikiTQ, FeTaQA, and WikiSQL datasets with improved robustness and efficiency compared to existing models.

Strengths:
1. The paper presents a novel and comprehensive framework for table QA, addressing the challenge of noise reduction through relevance scoring and parsing statements. It outperforms other tabular language models and GPT3-based methods, demonstrating state-of-the-art performance on various datasets.
2. The use of the ToTTo dataset for training a cell highlighter model adds a valuable contribution to the community, with ~300 manually annotated examples for parsing statement generation.

Weaknesses:
1. The paper lacks a clear rationale for how the different loss functions interact with each other in the proposed framework. A more thorough explanation of the interplay between these loss functions and their impact on model performance could enhance the paper's comprehensibility.
2. Unlike a "hard" table token retriever, the computation of the model proposed by this paper may be extensive when the number of table tokens is many.
3. Tabular dataset can be indexed and converted to their freeform text representations which general open-domain QA systems are already able to solve.

**Justification For Why Not Higher Score:**

While the weaknesses are not critical for the acceptance of the paper, it could indeed improve in presentation and more discussions around the model's complexity and interactions among loss functions.

**Justification For Why Not Lower Score:**

The paper introduces an innovative framework with promising results in table QA, and got unanimous ratings from all reviewers.

---

### Decision · Program_Chairs · 2024-01-16

Accept (spotlight)